# Bridging the Knowledge-Prediction Gap in LLMs on Multiple-Choice Questions

## Abstract

Large Language Models (LLMs) often fail on multiple-choice questions (MCQs) despite demonstrating correct knowledge in other contexts, such as free-form generation. To investigate the mechanism underlying this knowledge-prediction gap and alleviate it, we conduct a probing analysis on binary-choice questions and find that residual streams in certain layers contain a subspace spanned by two important bases: a *knowledge basis* that encodes the probability of the ground-truth answer and a *prediction basis* that encodes the probability of the answer choice predicted by the model. We observe that incorrect predictions arise from a misalignment of the model's hidden states along these two bases. Hence, we introduce **KAPPA** (Knowledge-Aligned Prediction through Projection-based Adjustment), an inference-time intervention that transforms hidden states to align the prediction coordinate with the knowledge coordinate. Experiments on binary-choice reformulations of Big-Bench-Hard show that KAPPA substantially improves accuracy and consistently outperforms baselines. KAPPA's benefit further extends to general MCQs, precisely mitigating the knowledge-predictino gap. Our work provides a new geometric understanding of the knowledge-prediction gap and offers a practical method for better aligning model behavior with its latent knowledge.[1]

## 1 Introduction

Multiple-choice questions (MCQs) are a widely used paradigm for evaluating large language models (LLMs) on knowledge- and reasoning-intensive tasks, as they provide clear ground truth and straightforward scoring. However, LLMs often fail to reliably predict correct answers in MCQ settings despite having the required knowledge. For instance, a model may generate the correct answer in a free-form setting but select the wrong option for the same question in the MCQ format (Figure 1a). Recent studies show that LLMs encode correct answers in their internal representations, but fail to consistently connect them to their output predictions (Marks & Tegmark, 2024; Liu et al., 2023a). Even when LLMs select an incorrect answer, the hidden representations frequently encode the correct answer, as detected by simple linear probes (Liu et al., 2023a; Marks & Tegmark, 2024; Bao et al., 2025). This knowledge-prediction gap is problematic because the model's knowledge is not fully exploited in its predictions, and the model exhibits unfaithful behavior where its outputs are driven by unpredictable mechanisms. Although this mismatch has been observed, its representational mechanism remains underexplored. In this work, we focus on MCQs and examine the hidden states of LLMs to better understand the knowledge-prediction gap and explore ways to mitigate it.

We first conduct a probing analysis of LLM representations on binary-choice questions. We find that in the residual streams of certain layers, there exists a subspace spanned by two functionally distinct bases: a *knowledge* basis that encodes the probability of the correct answer choice and a *prediction* basis that encodes the probability of the answer choice eventually predicted by the LLM. To construct this subspace, we train two logistic regression classifiers (Alain & Bengio, 2017) with disjoint objectives on the same hidden states: a knowledge probe that estimates how likely each option is to be correct, and a prediction probe that estimates how likely the LLM is to choose each option. Since the probe weight vectors define two distinct directions in the LLM's activation space, we span them to obtain a subspace. Projecting hidden states onto these bases yields coordinates

---

[1]We will release our code publicly upon publication.

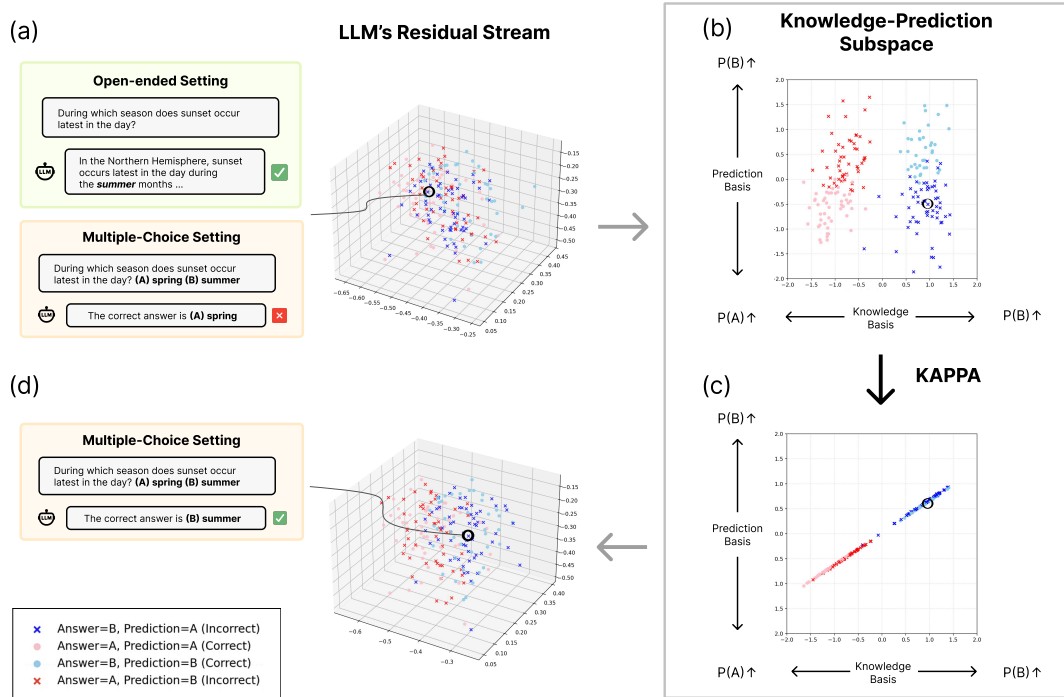

Figure 1: KAPPA resolves the knowledge-prediction gap through geometric realignment. The examples provided here use the binary-choice setting to facilitate intuitive understanding. (a) Motivating example: an LLM answers correctly in free-form but fails in MCQ format. 3D visualization shows hidden representations. (b) In the 2D knowledge-prediction subspace, the circled point shows misalignment between knowledge (x-axis) and prediction (y-axis) coordinates. (c) KAPPA geometrically transforms representations to align coordinates. (d) This correction enables faithful expression of internal knowledge, producing the correct answer.

corresponding to the ground-truth likelihood (knowledge) and the model's chosen answer (prediction). We observe that hidden state activations that lead to correct predictions often align well along both bases, whereas misalignments often result in incorrect predictions (Figure 1b). To extend our analysis to general $k$-option MCQs, we train knowledge and prediction probes as two single-layer $k$-class classifiers. These probes compute $k$-dimensional logits that estimate the likelihood that each option is the correct answer (knowledge) and the likelihood that the LLM will select each option (prediction), respectively. Empirically, LLMs often exhibit misalignment between the two logits.

To mitigate this mismatch, we introduce KAPPA (**K**nowledge–**A**ligned **P**rediction through **P**rojection-based **A**djustment), a test-time intervention that aligns the model's prediction with the knowledge encoded in its hidden representations. KAPPA (i) computes the knowledge and prediction coordinates of the test-time hidden states and (ii) minimally adjusts the hidden states so that the prediction coordinate matches the knowledge coordinate (Figure 1c). For general $k$-option MCQs, this procedure aligns the two $k$-dimensional logits by matching each hidden state's coordinates in the prediction subspace to those in the knowledge subspace, spanned by the corresponding probe weights. The update takes the form of an affine transformation within the prediction subspace. Through this simple and lightweight geometric correction, the model's choice becomes more faithful to the knowledge present in its representations.

Our experiments on multiple-choice tasks and free-form generation reveal three key findings about the knowledge-prediction gap and KAPPA's effectiveness: **(i) Simple geometric correction is sufficient to bridge the knowledge-prediction gap.** Despite operating only within a 2D subspace at a single layer, KAPPA consistently increases the agreement between the model's chosen answers and the knowledge probe, and effectively mitigates cases where the model generates the correct free-form answer but fails on the corresponding MCQ. These results indicate that the knowledge-prediction relationship is causally mediated by a low-dimensional geometric structure in the residual

stream. **(ii) A large portion of model failures arises from this gap.** On the challenging BBH benchmark, KAPPA substantially improves the performance of `Llama-2-7B-Chat` (Touvron et al., 2023): accuracy rises from near-chance (50.9%) to 70.8% on the binary reformulation. KAPPA further delivers a nearly 10% gain on the 4-choice reformulation. This substantial gain demonstrates that the model frequently hold the correct knowledge but fails to express it in its predictions. **(iii) The knowledge–prediction subspace captures generalizable structure.** Our analysis captures a fundamental geometric structure of the model's representations that extends across tasks and formats. Probes trained on one dataset (e.g., MMLU-Binary) transfer to others (e.g., BBH-Binary) with up to 4% accuracy gains, showing that the subspace encodes domain-general patterns. Moreover, applying KAPPA to free-form BBH improves accuracy by 3.8%, demonstrating that manipulating this subspace reliably elicits the model's encoded knowledge regardless of format.

**Contributions.** Our work provides a representation analysis of the knowledge-prediction gap in LLMs on MCQs and introduces a test-time intervention as a solution. Together, our research offers an interpretable framework for analyzing and manipulating the model's knowledge-prediction relationship. Our contributions are threefold:

- We empirically demonstrate the knowledge-prediction gap using linear probes that separately capture the model's encoded knowledge and its actual answer choices for MCQs.
- We provide a geometric characterization of this gap by presenting a method to identify an interpretable knowledge-prediction subspace from the residual stream.
- We introduce a closed-form inference-time intervention that calibrates hidden states within this subspace, and show that this lightweight intervention reliably aligns the model's predictions with its internal knowledge.

## 2 RELATED WORK

**The Knowledge-Prediction Gap.** Recent studies have highlighted the mismatch between LLMs' internal knowledge and their external predictions. Prior work shows that linear probes can extract factual correctness of model output from hidden representations even when the model outputs are wrong (Marks & Tegmark, 2024; Liu et al., 2023a; Azaria & Mitchell, 2023; Wang et al., 2024; Su et al., 2024; Orgad et al., 2025). Several explanations have been proposed, including: (i) distractor-driven mechanisms that overrides correct knowledge (Wiegreffe et al., 2025; Yang et al., 2025; Tulchinskii et al., 2024) and (ii) miscalibration between early knowledge-encoding layers and later prediction-generating layers (Gottesman & Geva, 2024). In this work, we investigate a geometric structure of the hidden representations that offers a more interpretable view on the phenomenon.

**Inference-time Intervention.** Recent work has increasingly explored modifying model behavior through representation-level interventions (Zou et al., 2025; Arditi et al., 2024; Rimsky et al., 2024; Hendel et al., 2023; Ilharco et al., 2023). Our approach methodologically follows the inference-time intervention paradigm (Li et al., 2023), where prior work similarly trains linear probes to identify meaningful directions in hidden space and then modifies activations during inference (Li et al., 2023; von Rütte et al., 2024; Chen et al., 2024). This line of research has been applied to a wide range of objectives, including toxicity mitigation (Turner et al., 2023; Liu et al., 2023b), safety (Zou et al., 2023; Rimsky et al., 2024; Lee et al., 2025), instruction-following (Stolfo et al., 2025; Heo et al., 2025), and even reasoning tasks (Venhoff et al., 2025).

**Differences with Relevant Approaches.** Unlike typical **activation steering** methods that applies a fixed vector across all hidden states (Turner et al., 2023; Rimsky et al., 2024), KAPPA applies instance-specific updates to hidden states dynamically. Unlike **Fine-tuning** methods that modify LLM parameters to introduce new task-specific information (Wei et al., 2022; Ouyang et al., 2022; Hu et al., 2022), KAPPA leaves all LLM parameters unchanged. Instead, KAPPA uses lightweight probes to identify a task-relevant subspace and intervenes only at inference time.

## 3 METHOD

We present KAPPA, a test-time intervention that mitigates the knowledge-prediction gap by applying transformations to residual streams. Our pipeline proceeds in three steps: we collect residual stream

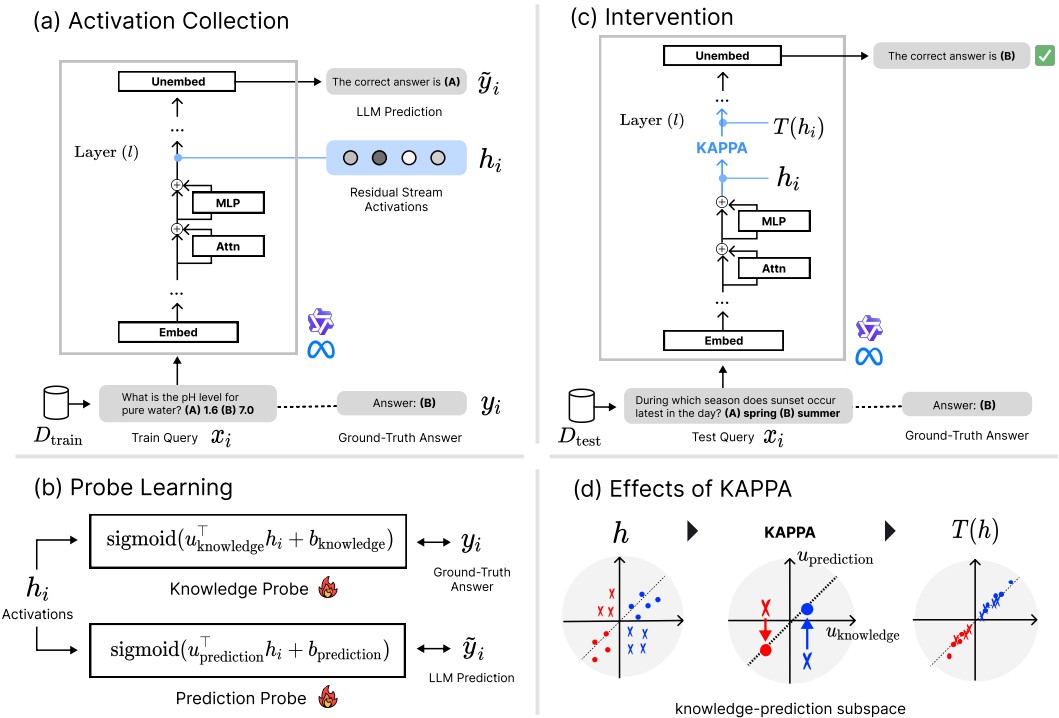

Figure 2: KAPPA pipeline overview. (1) Collect residual stream activations from transformer layers during inference on binary-choice questions. (2) Train knowledge and prediction probes: logistic regression classifiers that predict ground-truth labels and model outputs, respectively, from the same hidden states. (3) At inference, KAPPA applies a geometric transformation to the hidden representation. (4) 2D visualization showing the distribution of representations before (left) and after (right) the intervention in the knowledge-prediction subspace.

activations from MCQs (Figure 2a), identify the knowledge-prediction subspace via linear probes (Figure 2b), and apply intervention within the subspace (Figure 2c). We first describe the KAPPA methodology formulated for binary-choice questions (§ 3.1, § 3.2) and subsequently generalize it to $k$-option MCQs (§ 3.3).

### 3.1 IDENTIFYING KNOWLEDGE AND PREDICTION BASES

Our intervention operates within a subspace of the residual stream that captures the knowledge-prediction relationship. For each layer, we define a subspace spanned by a knowledge basis, which encodes the probability of the correct answer choice, and a prediction basis, which encodes the probability of the model's own choice. The bases are obtained by training linear probes.

**Dataset Construction.** To learn the bases, we construct training datasets for linear probes. For each input prompt $x_i \in D_{\text{train}}$ containing a binary-choice question, we store both the ground-truth label $y_i \in \mathbb{R}$ and the model's prediction $\tilde{y}_i \in \mathbb{R}$. At every layer $l$, we extract the residual stream activation of the last token of $x_i$, denoted as $h^{(l)}(x_i) \in \mathbb{R}^d$, and construct two datasets. The *knowledge* dataset, $D_{\text{knowledge}}^{(l)} = \{(h^{(l)}(x_i), y_i)\}$, pairs activations with ground-truth labels, while the *prediction* dataset, $D_{\text{prediction}}^{(l)} = \{(h^{(l)}(x_i), \tilde{y}_i)\}$, pairs the same activations with the model's predicted choices.

**Linear Probing.** We train separate logistic regression classifiers on each dataset for each layer $l$. Concretely, we train two linear classifiers with distinct objectives: **(i) Knowledge probe**, a classifier trained to predict the ground-truth label $y_i$ from $h^{(l)}(x_i)$, and **(ii) Prediction probe**, a classifier trained to predict the model's output label $\tilde{y}_i$ from $h^{(l)}(x_i)$. Each probe has parameters consisting of a weight vector $u^{(l)} \in \mathbb{R}^d$ and a bias term $b^{(l)}$. Since the probe weight vectors $u_{\text{knowledge}}^{(l)}$ and $u_{\text{prediction}}^{(l)}$ define two distinct directions in the activation space, we use them as basis vectors to

span the knowledge–prediction subspace. Any hidden state $h^{(l)}(x_i)$ can be projected into this 2D subspace by computing its coordinates along the two bases:

$$\text{Knowledge coordinate: } u_{\text{knowledge}}^{(l)\top}h^{(l)}(x_i), \qquad \text{Prediction coordinate: } u_{\text{prediction}}^{(l)\top}h^{(l)}(x_i).$$

These quantities correspond to the logits produced by the linear probes: For the knowledge basis, a larger coordinate indicates that the hidden states assign a higher probability to the option being a correct answer. Similarly, a larger coordinate on the prediction basis reflects a higher probability of the option being chosen by the model.

As shown in Figure 2d, this construction provides an interpretable two-dimensional geometry that captures the relationship between the model's internal knowledge and its external predictions. Ideally, an option that is more likely to be correct should also be the option the model is more likely to choose. In practice, however, we frequently observe mismatches between the two coordinates, which motivates a geometric correction that explicitly aligns them.

## 3.2 KNOWLEDGE-ALIGNED PREDICTION ADJUSTMENT

At inference time, we calibrate the two coordinates of the hidden state within the subspace. Given a question $x_i \in D_{\text{test}}$, we intervene at a chosen layer $l$ by applying a transformation to the residual stream activation $h = h^{(l)}(x_i)$. For clarity, we omit the layer index $(l)$ in the following notation.

**Transformation Objective.** Our goal is to derive a transformation that aligns the prediction coordinate with the corresponding knowledge coordinate of each hidden state. For a hidden state $h$, our intervention computes a transformation $\mathcal{T} : \mathbb{R}^d \to \mathbb{R}^d$ that maps $h$ to a modified state $h' = \mathcal{T}(h)$ under the following design: (1) *Alignment constraint*: the modified prediction coordinate must align with the knowledge coordinate; (2) *Minimal Perturbation*: the perturbation $\|h' - h\|$ is minimized so as to preserve other encoded information.

**Constrained Optimization.** We formulate the update as a constrained optimization problem:

$$\min_{h'} \quad \|h' - h\|_2^2 \quad \text{s.t.} \quad \tilde{u}_{\text{prediction}}^{\top}\tilde{h}' = \tilde{u}_{\text{knowledge}}^{\top}\tilde{h}, \tag{1}$$

where we use augmented notations

$$\tilde{h}' = \begin{bmatrix} h' \\ 1 \end{bmatrix}, \quad \tilde{h} = \begin{bmatrix} h \\ 1 \end{bmatrix}, \quad \tilde{u}_{\text{knowledge}} = \begin{bmatrix} u_{\text{knowledge}} \\ b_{\text{knowledge}} \end{bmatrix}, \quad \tilde{u}_{\text{prediction}} = \begin{bmatrix} u_{\text{prediction}} \\ b_{\text{prediction}} \end{bmatrix} \in \mathbb{R}^{d+1}.$$

Solving this problem yields the minimal $\ell_2$ modification to $h$ that satisfies the alignment constraint. We derive a closed-form solution, which takes the form of a rank-1 affine transformation of $h$:

$$h' = h + u_{\text{prediction}} \frac{\tilde{u}_{\text{knowledge}}^{\top}\tilde{h} - \tilde{u}_{\text{prediction}}^{\top}\tilde{h}}{\|u_{\text{prediction}}\|^2}. \tag{2}$$

This transformation modifies the hidden state only in the prediction basis $u_{\text{prediction}}$, leaving all orthogonal components unchanged (Figure 2d). In contrast to typical activation steering methods that apply fixed modifications across all activations, our approach dynamically computes the minimal adjustment required based on the knowledge-prediction coordinates. The full derivation, interpretations, and comparisons to LoRA updates are provided in Appendix B.

**Generalized Alignment.** While the original alignment constraint in Eq. 1 enforces strict equality between the knowledge and prediction coordinates, we extend our method to a more general form to capture more diverse knowledge-prediction relationships. We parameterize the target mapping with hyperparameters $w \in \mathbb{R}$ and $\beta \in \mathbb{R}$:

$$\tilde{u}_{\text{prediction}}^{\top}\tilde{h}' = w \cdot \tilde{u}_{\text{knowledge}}^{\top}\tilde{h} + \beta \cdot \text{sign}(\tilde{u}_{\text{knowledge}}^{\top}\tilde{h}). \tag{3}$$

The original constraint is recovered when $w = 1$ and $\beta = 0$, while larger values of $w$ and $\beta$ amplify the influence of internal knowledge on model predictions. This generalized formulation accommodates a broader range of relationships, with optimal $(w, \beta)$ depending on the task. To provide a comprehensive understanding of our method, we visualize the resulting transformations of hidden states in Appendix D and analyze the parameter effects in §4.6.

### 3.3 EXTENSION TO MULTIPLE-CHOICE SETTINGS

We extend our method from binary to $k$-option MCQs by identifying geometric directions that encode how likely each option is correct (knowledge) and how likely the model is to choose it (prediction), and then aligning hidden states along the directions.

**Probing.** For a $k$-option MCQ, we collect residual stream activations and we train the knowledge and prediction probes as two single-layer $k$-class classifiers. Specifically, given a hidden state $h \in \mathbb{R}^{d \times 1}$, $U_{\text{knowledge}}^{\top} h + b_{\text{knowledge}}, U_{\text{prediction}}^{\top} h + b_{\text{prediction}} \in \mathbb{R}^{k}$, where $U_{\text{knowledge}}, U_{\text{prediction}} \in \mathbb{R}^{d \times k}$ are weight matrices and $b_{\text{knowledge}}, b_{\text{prediction}} \in \mathbb{R}^{k \times 1}$ are biases, represent the logits for how likely each option is to be correct (knowledge) and how likely the model is to choose each option (prediction), respectively. These logits can be interpreted as coordinates in two $k$-dimensional subspaces of the residual stream: the knowledge subspace $\text{span}(U_{\text{knowledge}})$ and the prediction subspace $\text{span}(U_{\text{prediction}})$.

**Intervention.** Following the same principle as scalar coordinate alignment in the binary case, the multiple-choice setting extends this procedure by aligning the $k$-dimensional logits produced by the knowledge and prediction probes. This can be interpreted as aligning each hidden state's coordinates in the prediction subspace with those in the knowledge subspace. To calibrate hidden states jointly along all k options, we solve the generalized version of the constrained optimization problem as the binary case. This yields the generalized affine update:

$$\tilde{h}' = \tilde{h} + \tilde{U}_{\text{prediction}} \left( \tilde{U}_{\text{prediction}}^{\top} \tilde{U}_{\text{prediction}} \right)^{-1} \left( \tilde{U}_{\text{knowledge}}^{\top} \tilde{h} - \tilde{U}_{\text{prediction}}^{\top} \tilde{h} \right),$$

which reduces to the binary case when $k = 1$. The update modifies the representation only within the prediction subspace, adjusting it to match the geometry of the representation in the knowledge subspace. A detailed formulation and full derivation are provided in Appendix C.

## 4 EXPERIMENTS

We evaluate probe reliability (§4.1) and demonstrate KAPPA's effectiveness on binary-choice benchmarks, including cross-dataset transfer (§4.2). We then assess the generalized KAPPA on 3- and 4-choice benchmarks (§4.3). Next, we show KAPPA resolves the gaps in free-form generation and assess effects on general capabilities (§4.4, §4.5). Finally, we analyze hyperparameter effects (§4.6).

To evaluate our method on reasoning- and knowledge-intensive tasks, we use subsets from three widely adopted MCQ benchmarks: Big-Bench Hard (BBH) (Suzgun et al., 2023), Massive Multi-task Language Understanding (MMLU) (Hendrycks et al., 2021), and AI2 Reasoning Challenge challenge set (ARC-Challenge) (Clark et al., 2018). The full list of subsets and data statistics is provided in Appendix E. As a first step, we convert each MCQ item into a binary-choice format by pairing the correct answer with a randomly selected distractor, yielding BBH-Binary, MMLU-Binary, and ARC-Challenge-Binary, and evaluate KAPPA on binary-choice settings. We also evaluate our method on standard multi-option settings using the 4-choice formulation of BBH (BBH-4C), the original 4-option MMLU (MMLU-4C), and the original 3-option ARC-Challenge (ARC-Challenge-3C) (§ 4.3). We primarily evaluate on `Llama-2-7B-Chat` (Touvron et al., 2023) and `Qwen2.5-7B-Instruct` (Qwen et al., 2025), denoted as `Llama-2` and `Qwen-2.5`, to demonstrate the broad applicability of KAPPA across architectures.

### 4.1 LINEAR PROBE EVALUATION

We report the test performance of our trained probes on binary-choice datasets in Figure 3. Our analysis yields two key insights: First, linear probes reliably capture knowledge and prediction signals from hidden states in the middle-to-late layers, consistently achieving over 70% accuracy and in some cases exceeding 95%. Knowledge probe accuracy is low in early layers but rises in mid-to-late layers, with variation across models yet consistent layer patterns across datasets, aligning with prior work showing that LLMs encode richer semantic and knowledge representations in later layers (Skean et al., 2025; Meng et al., 2022; Dai et al., 2022). Second, knowledge probes often surpass the model's own prediction accuracy. This occurs in at least one layer for every setting shown in Figure 3. In particular, on BBH-Binary with the `Llama-2`, the knowledge probe achieves over 70% accuracy, while the base model itself performs near chance level (50.9%). These findings in-

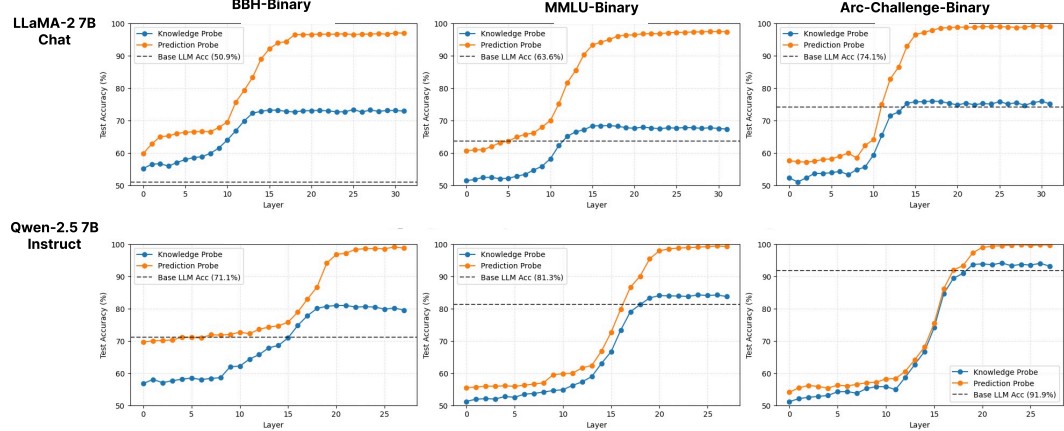

Figure 3: Accuracy of knowledge probes (blue) and prediction probes (orange) across transformer layers for `LLaMA-2` and `Qwen-2.5` on BBH-Binary, MMLU-Binary, and ARC-Challenge-Binary datasets. Dashed lines indicate base model accuracy for each dataset.

| Method | Llama-2-7B-Chat | | | Qwen2.5-7B-Instruct | | |
|---|---|---|---|---|---|---|
| | BBH | MMLU | ARC-Challenge | BBH | MMLU | ARC-Challenge |
| Base | 50.9 | 63.6 | 74.1 | 71.1 | 80.8 | 91.9 |
| LoRA FT | 79.2 | 75.6 | 82.0 | 86.1 | 84.6 | 94.6 |
| ICL (BM25) | 51.4 | 65.1 | 75.8 | 71.0 | 84.7 | 95.2 |
| Steering | 50.1 | 63.3 | 74.2 | 71.0 | 81.6 | 92.0 |
| KAPPA (1-layer) | 69.8 | 64.9 | 76.3 | 72.3 | 83.0 | 93.1 |
| KAPPA (6-layer) | 70.8 | 66.2 | 75.8 | 74.3 | 83.3 | 93.5 |
| Knowledge Probe | 72.8 | 68.4 | 76.0 | 81.0 | 84.1 | 94.0 |

Table 1: Results of KAPPA intervention on binary-choice reformulations of BBH, MMLU, and ARC-Challenge datasets using `LLaMA-2` and `Qwen-2.5`. Numbers represent test accuracy percentages. KAPPA variants indicate interventions applied to 1 layer or 6 layers, respectively.

dicate that models internally encode correct knowledge even when their predictions are incorrect, highlighting the significance of the knowledge-prediction gap.

## 4.2 BINARY-CHOICE INTERVENTION

**Experiment Setup.** Using the knowledge and prediction probes trained on each layer for each dataset, we KAPPA at inference time. For analysis on generalization, we conduct a cross-dataset transfer experiment, where the knowledge-prediction subspace is identified using one dataset and then directly applied to another. This setup tests whether the subspaces are shared and reusable across datasets. Details on intervention layers and hyperparameters are provided in Appendix F.1.

**Baselines.** To illustrate the practical performance achieved by our method, we compare our method against the following baselines: (i) **Base**: the original LLM without any intervention. (ii) **Steering**: activation steering using logistic regression weights (Li et al., 2023). (iii) **LoRA FT** (Hu et al., 2022): LoRA fine-tuning on the train dataset. (iv) **ICL (BM25)** (Brown et al., 2020): in-context learning (ICL) with BM25-retrieved demonstrations. As our method is an inference-time intervention, the primary baseline is activation steering. More details are presented in Appendix F.2

**Results.** Table 1 demonstrates KAPPA's effectiveness across models and tasks. KAPPA consistently and reliably improves the base model across all evaluated settings. We report average performance across subsets within each dataset and individual results are provided in Appendix J.

**Aligning Model Predictions with Knowledge Probe Outputs.** Since our primary goal is to ensure that the knowledge encoded along the knowledge basis is faithfully reflected in the model's predictions, the natural performance target is the accuracy of the knowledge probe. Because the probe is trained on frozen representations, it measures the linearly extractable knowledge already

| Method | Test Data / Train Data | Llama-2-7B-Chat | | | Qwen2.5-7B-Instruct | | |
|---|---|---|---|---|---|---|---|
| | | BBH | MMLU | ARC-Challenge | BBH | MMLU | ARC-Challenge |
| Base | - | 50.9 | 63.6 | 74.1 | 71.1 | 81.3 | 91.9 |
| ICL | BBH | 51.4 | 66.0 | 73.5 | 71.0 | 83.5 | 94.3 |
| | MMLU | 52.6 | 65.1 | 74.4 | 69.5 | 84.7 | 95.0 |
| | ARC-Challenge | 51.5 | 66.9 | 75.8 | 69.2 | 83.4 | 95.2 |
| LoRA FT | BBH | 79.2 | 67.8 | 75.9 | 86.1 | 82.8 | 94.6 |
| | MMLU | 57.8 | 75.6 | 79.8 | 82.8 | 84.6 | 95.3 |
| | ARC-Challenge | 54.0 | 71.0 | 82.0 | 76.0 | 82.6 | 94.6 |
| Steering | BBH | 50.1 | 63.6 | 74.6 | 71.0 | 81.6 | 91.9 |
| | MMLU | 50.3 | 63.3 | 74.6 | 71.2 | 81.6 | 91.9 |
| | ARC-Challenge | 49.6 | 63.2 | 74.2 | 71.1 | 81.4 | 92.0 |
| KAPPA (1-layer) | BBH | 69.8 | 63.2 | 73.3 | 72.3 | 81.0 | 91.9 |
| | MMLU | 54.0 | 64.9 | 73.2 | 71.6 | 83.0 | 93.5 |
| | ARC-Challenge | 54.9 | 63.2 | 76.3 | 72.0 | 82.4 | 93.1 |
| KAPPA (6-layers) | BBH | 70.8 | 61.6 | 71.6 | 74.3 | 80.6 | 91.8 |
| | MMLU | 54.0 | 66.2 | 75.5 | 71.1 | 83.3 | 93.6 |
| | ARC-Challenge | 52.5 | 62.9 | 75.8 | 68.6 | 81.3 | 93.5 |

Table 2: Cross-dataset generalization results for KAPPA and baseline methods. Each method is trained using one dataset (rows) and applied to different test datasets (columns). Diagonal entries show in-domain performance; off-diagonal entries demonstrate cross-dataset transfer capability. - Binary versions are used for all datasets.

present in the model's parameters. KAPPA consistently closes the gap to this knowledge probe performance (row 2 of Table 1) and makes the model's answer choices substantially more aligned with the knowledge probe's predictions. As shown in Table 8, the agreement rate between the probe's predicted answers and the model's outputs increases significantly after applying KAPPA.

**Substantial improvements on reasoning-intensive tasks.** BBH-Binary is a particularly challenging benchmark that demands sophisticated reasoning, as reflected in `Llama-2`'s near-random baseline performance (50.9%). KAPPA achieves a remarkable 18.9% point improvement, substantially surpassing comparable inference-time methods—activation steering (+20.7%) and ICL (+18.4%). Even for stronger models like Qwen-2.5, KAPPA still provides 1–3% gains, indicating that knowledge–prediction gaps persist across models. These results show that models often encode the knowledge yet fail to express it in their predictions, which KAPPA effectively mitigates.

**Consistent gains across knowledge-intensive domains.** On knowledge-heavy tasks (MMLU-Binary, ARC-Challenge-Binary), KAPPA shows steady 1-3% point improvements across models. Notably, KAPPA achieves approximately 80-90% of LoRA's performance despite utilizing only a single-layer intervention with 32×16×4 times fewer additional parameters (Llama-2). This result supports for our hypothesis: a substantial portion of the model's errors stems from geometric misalignment between knowledge and prediction rather than insufficient knowledge capacity.

**Generalization across diverse domains and tasks.** KAPPA demonstrates cross-dataset generalization (Table 2), revealing that the knowledge-prediction subspace captures universal aspects of the internal knowledge rather than task-specific artifacts. Probes trained on MMLU and ARC-Challenge transfer effectively to BBH (Binary versions), improving `Llama-2` accuracy by up to 4% points and surpassing steering by over 2% points. Even for the stronger `Qwen-2.5`, MMLU-trained probes improve ARC-Challenge accuracy by nearly 2% points. Across all datasets, KAPPA preserves performance without notable degradation, underscoring the robustness of the underlying geometry.

**KAPPA Improves Free-Form–MCQ Mismatched Cases.** We hypothesize that when models answer correctly in free-form but fail in MCQs, they possess the knowledge but cannot effectively express it for the MCQs. As shown in Figure 4a, `Llama-2` exhibits this inconsistency in 11.4% of cases, and 5.7% remain unresolved even after LoRA fine-tuning. KAPPA further reduces this gap to 5.1%, indicating its precision in addressing the specific mismatch. This result indicates that KAPPA directly targets and effectively mitigates the underlying knowledge–prediction gap.

**Improvements Across a Broad Set of Models.** We evaluated our approach on a broader range of models, including `Qwen3-32B`, `Llama-3.1-8B`, and `Qwen3-4B` on the BBH-binary dataset. As shown in Table 16, all three models exhibit a persistent knowledge-prediction gap, and KAPPA improves their accuracies by +9.1%, +13.3%, and +13.7%, respectively. These results suggest that the gap is pervasive across model families and sizes, and that KAPPA consistently mitigates it.

|  | Llama-2-7B-Chat | | | Qwen2.5-7B-Instruct | | |
|--------|------|------|------|------|------|------|
| **Method** | BBH | MMLU | ARC | BBH | MMLU | ARC |
| Base | 27.6 | 38.8 | 62.5 | 53.0 | 69.3 | 89.5 |
| KAPPA (1-layer) | 34.1 | 43.9 | 64.4 | 56.1 | 69.9 | 89.7 |
| KAPPA (3-layer) | 36.9 | 44.5 | 63.8 | 55.4 | 69.9 | 89.6 |
| Knowledge Probe | 44.4 | 45.2 | 64.5 | 57.7 | 70.3 | 89.7 |

Table 3: Results of Multiple-Choice Intervention on BBH-4C, MMLU-4C, and ARC-3C using `Llama-2` and `Qwen-2.5`. Numbers represent test accuracy percentages. KAPPA variants correspond to interventions applied to 1, 3 layers, respectively.

Taken together, the results consistently support our hypothesis: LLMs often encode correct answers internally yet fail to express this knowledge in their predictions. Applying simple calibration through KAPPA substantially improves performance, revealing two key findings: (1) the prediction basis has a causal effect on model behavior, and (2) modifying the hidden representation only within this 2D subspace is sufficient to make the model's predictions reflect its latent knowledge. These results indicate that a substantial portion of model failures can be attributed to the knowledge–prediction gap, and the gap is causally mediated by a simple 2D subspace in the model's representation.

### 4.3 MULTIPLE-CHOICE INTERVENTION

We evaluate the generalized KAPPA (§ 3.3) and find that KAPPA generalizes well to multiple-choice settings. As shown in Table 3, the `Llama-2` base model exhibits a substantial gap between its prediction accuracy and the knowledge probe's accuracy, and KAPPA consistently narrows this gap even under the more complex multi-option geometry. Notably, KAPPA improves BBH-4C accuracy by nearly +10% and MMLU-4C accuracy by nearly +5%. For `Qwen-2.5`, the gains are smaller—reflecting its initially smaller knowledge-prediction gap—but KAPPA further reducees the discrepancy. Overall, these results show that while the magnitude of the knowledge-prediction gap varies across models, KAPPA consistently reduces this gap even in challenging multiple-choice settings.

### 4.4 FREE-FORM GENERATION

**Experimental Setup.** We test whether KAPPA's benefits can be extended to improve free-form generation. As detailed in Appendix M, we construct a free-form dataset from BBH by systematically converting MCQ items into free-form questions. This allows us to reuse the already-characterized knowledge-prediction subspace. We apply KAPPA, trained on BBH-Binary, to free-form generation by applying the intervention at every token-level decoding step. To assess the factual accuracy of the generated text, we use `gpt-4o-mini-2025-04-16` (OpenAI, 2024) to judge whether the generated response semantically includes or matches the ground-truth answer.

**Results.** We find that KAPPA's benefits extend beyond MCQs to improve free-form generation. Figure 4b shows that applying KAPPA increases the accuracy of free-form answers from 25.0% to 29.1%. This shows that KAPPA is not limited to specific answer formats but functions as a general intervention that adjusts the model's internal representations to better reflect its latent knowledge. Consequently, KAPPA shows the potential to close the knowledge–prediction gap across a broad range of generative tasks that demand accurate knowledge retrieval and reasoning.

### 4.5 IMPACT ON GENERAL CAPABILITY

**Experimental Setup.** We evaluate how KAPPA affects the general capabilities using `vicuna-eval` (Chiang et al., 2023), comparing against Steering and LoRA FT. For KAPPA and Steering, we apply probes trained on the BBH-Binary to the `Llama-2` model. Response quality is assessed with `gpt-5` (reasoning effort: high) (OpenAI, 2025), which compares outputs from KAPPA and the baselines, repeating each comparison with swapped response orders. Dataset details and evaluation prompts are provided in Appendix N.

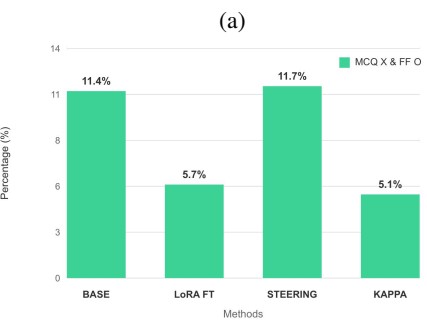
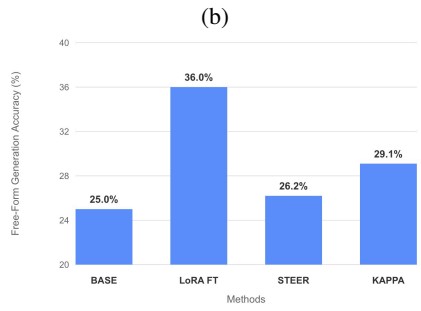

Figure 4: (a) Ratio of "MCQ wrong, free-form-correct" cases for each method. (b) Performance of each method on free-form version of BBH dataset.

**Result.** We find that KAPPA enhances performance on knowledge- and reasoning-intensive tasks without degrading the model's ability to generate coherent responses. It performs on par with the baselines and slightly surpasses steering. Full results are reported in Table 15. Notably, KAPPA attains win rates above 50% on subsets requiring factual retrieval and structured reasoning, highlighting its ability to more faithfully leverage internal knowledge during generation.

### 4.6 HYPERPARAMETER ANALYSIS

To examine the causal effects of the alignment parameters $w$ and $\beta$ in Equation 3, we conduct a hyperparameter sweep. Probes are trained and also evaluated on BBH-Binary. Our analysis shows that varying $w$ and $\beta$ produces clear causal effects: larger values consistently improve accuracy, narrowing the gap to the knowledge probe (Figure 5). Increasing either parameter amplifies the prediction coordinate, thereby steering outputs toward the knowledge probe's choice. These findings demonstrate that the hyperparameters causally steer the model to better match its internal knowledge.

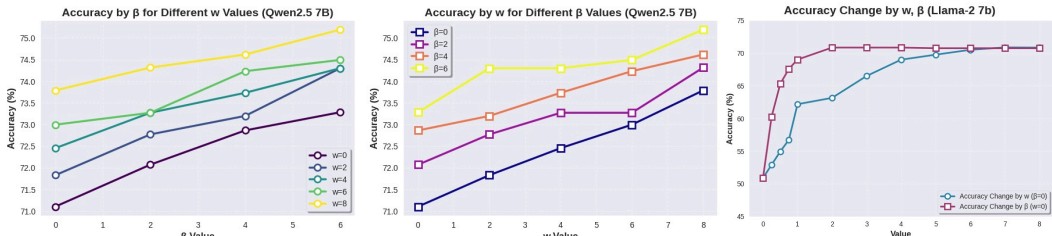

Figure 5: Effect of alignment parameters $w$ and $\beta$ on model accuracy. (**Left, Middle**) Qwen-2.5 7B Instruct results across five $w$ values (0, 2, 4, 6, 8) and four $\beta$ values (0, 2, 4, 6). (**Right**) Llama-2 7B Chat results when varying only $w$ ($\beta = 0$) versus only $\beta$ ($w = 0$). Across both models, increasing either parameter consistently boosts accuracy.

## 5 CONCLUSION

This work addresses the *knowledge-prediction gap* in LLMs—the phenomenon where models internally encode correct knowledge yet fail to express it in their predictions. We show that this gap is captured by a subspace of residual streams, spanned by a *knowledge basis* encoding ground-truth probabilities and a *prediction basis* encoding output probabilities. Our proposed method, KAPPA calibrates activations along these bases at inference time. Across binary-choice and multiple-choice settings, KAPPA yields substantial gains and shows consistent improvements. This lightweight calibration aligns the model's predicted choice with the answer encoded along the knowledge basis and precisely mitigates the knowledge–prediction gap. These findings offer a new geometric perspective on model failures and a practical method for eliciting more faithful expressions of internal knowledge. Future work will explore broader applications of KAPPA and further investigate the underlying causes of the knowledge–prediction gap.

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

# A  LLM Acknowledgment

We use LLM to polish the writing and refactor our codebase.

# B  Optimized Transformation of KAPPA

## B.1  Derivation of the Closed-form Update

We rewrite our optimization problem Eq. 1 as

$$\min_{h'} \quad \|h' - h\|_2^2 \quad \text{s.t.} \quad u_{\text{prediction}}^\top h' + b_{\text{prediction}} = u_{\text{knowledge}}^\top h + b_{\text{knowledge}}. \tag{4}$$

Equivalently, the constraint can be written as

$$u_{\text{prediction}}^\top h' = p_{\text{target}}, \quad p_{\text{target}} := u_{\text{knowledge}}^\top h + b_{\text{knowledge}} - b_{\text{prediction}}.$$

We derive the solution via Lagrange multipliers. Define the Lagrangian

$$\mathcal{L}(h', \lambda) = \|h' - h\|_2^2 + \lambda\big(u_{\text{prediction}}^\top h' - p_{\text{target}}\big),$$

with multiplier $\lambda \in \mathbb{R}$. Setting the gradient with respect to $h'$ to zero gives

$$2(h' - h) + \lambda u_{\text{prediction}} = 0 \quad \Longrightarrow \quad h' = h - \tfrac{\lambda}{2} u_{\text{prediction}}.$$

Substituting this into the constraint,

$$u_{\text{prediction}}^\top h - \tfrac{\lambda}{2} u_{\text{prediction}}^\top u_{\text{prediction}} = p_{\text{target}},$$

yields

$$\lambda = 2 \, \frac{u_{\text{prediction}}^\top h - p_{\text{target}}}{u_{\text{prediction}}^\top u_{\text{prediction}}}.$$

Hence, the closed-form solution is

$$h' = h + u_{\text{prediction}} \frac{p_{\text{target}} - u_{\text{prediction}}^\top h}{u_{\text{prediction}}^\top u_{\text{prediction}}}.$$

This coincides with the derived solution in Eq. 2, after resolving $p_{\text{target}}$.

## B.2  Interpretations of the Closed-form Update

The solution yields the closed-form update, which is an affine transformation of $h$:

$$h' = \big(I - \frac{u_{\text{prediction}} u_{\text{prediction}}^\top}{\|u_{\text{prediction}}\|^2}\big)h + \frac{p_{\text{target}}}{\|u_{\text{prediction}}\|^2} u_{\text{prediction}}.$$

Note that the original hidden state $h$ can be decomposed as

$$h = \Big(I - \frac{u_{\text{prediction}} u_{\text{prediction}}^\top}{\|u_{\text{prediction}}\|^2}\Big)h + \frac{u_{\text{prediction}}^\top h}{\|u_{\text{prediction}}\|^2} u_{\text{prediction}}.$$

Compared to this decomposition, our update $h'$ simply replaces the prediction coordinate $\frac{u_{\text{prediction}}^\top h}{\|u_{\text{prediction}}\|^2}$ with $\frac{p_{\text{target}}}{\|u_{\text{prediction}}\|^2}$, while leaving all orthogonal components unchanged. Geometrically, $h'$ is the orthogonal projection of $h$ onto the hyperplane $\{z \mid u_{\text{prediction}}^\top z = p_{\text{target}}\}$.

**Comparisons to LoRA.** We can also rewrite the transformation as $h' = h + BAh + C$, where

$$B = \frac{u_{\text{pred}}}{u_{\text{pred}}^\top u_{\text{pred}}} \in \mathbb{R}^{d \times 1}, A = (u_{\text{know}} - u_{\text{pred}})^\top \in \mathbb{R}^{1 \times d}, C = u_{\text{pred}} \cdot \frac{b_{\text{know}} - b_{\text{pred}}}{u_{\text{pred}}^\top u_{\text{pred}}} \in \mathbb{R}^{d \times 1}.$$

Here, the matrix $(BA)$ is rank-1, and the bias term $(C)$ is typically very small in practice. Thus, the resulting affine transformation resembles a rank-1 LoRA update in form.

Despite KAPPA's similar form to LoRA, we further clarify below how KAPPA differs from LoRA in its operation.

- Input/output: LoRA adapters receive the input to each linear layer and modify the layer's output. In contrast, KAPPA takes the residual-stream activation as input and applies a transformation directly to the hidden state at inference time.

- Mechanism: LoRA adds a low-rank update to the existing weight matrix, altering model parameters through training. KAPPA leaves all weights untouched and applies an inference-time correction to the residual-stream activations.

- Parameter scale and role: The LoRA baseline attaches adapters to every linear weight across all layers to compress new task-specific information. KAPPA intervenes on only a single layer, thus using much fewer parameters. These parameters serve solely to ensure that the model's existing knowledge is faithfully expressed in its predictions, not to encode new knowledge.

## C EXTENSION METHOD FOR GENERAL MULTIPLE-CHOICE QUESTIONS

### C.1 CONSTRAINED OPTIMIZATION PROBLEM

We generalize the binary-choice derivation of KAPPA—which operates on $u_{\text{prediction}}, u_{\text{knowledge}} \in \mathbb{R}^d$—to the setting where the prediction and knowledge directions span $k$-dimensional subspaces:

$$U_{\text{prediction}}, U_{\text{knowledge}} \in \mathbb{R}^{d \times k}, \qquad k \geq 1.$$

We formulate the inference-time update over an augmented hidden state as the following constrained optimization:

$$\min_{\tilde{h}'} \quad \|\tilde{h}' - \tilde{h}\|_2^2 \qquad \text{s.t.} \qquad \tilde{U}_{\text{prediction}}^\top \tilde{h}' = \tilde{U}_{\text{knowledge}}^\top \tilde{h},$$

where we use augmented notations

$$\tilde{h}' = \begin{bmatrix} h' \\ 1 \end{bmatrix}, \quad \tilde{h} = \begin{bmatrix} h \\ 1 \end{bmatrix}, \qquad \tilde{U}_{\text{knowledge}} = \begin{bmatrix} U_{\text{knowledge}} \\ b_{\text{knowledge}}^\top \end{bmatrix}, \quad \tilde{U}_{\text{prediction}} = \begin{bmatrix} U_{\text{prediction}} \\ b_{\text{prediction}}^\top \end{bmatrix} \in \mathbb{R}^{(d+1) \times k}.$$

Solving this least-squares problem yields the closed-form update

$$\tilde{h}' = \tilde{h} + \tilde{U}_{\text{prediction}} \left( \tilde{U}_{\text{prediction}}^\top \tilde{U}_{\text{prediction}} \right)^{-1} \left( \tilde{U}_{\text{knowledge}}^\top \tilde{h} - \tilde{U}_{\text{prediction}}^\top \tilde{h} \right),$$

which is an affine transformation of $h$ that applies the minimal $\ell_2$ modification to satisfy the alignment constraint. Crucially, the update alters $h$ only within the prediction subspace $\text{span}(U\text{prediction})$ while leaving orthogonal components untouched. This formulation directly generalizes the binary case (§ 3.2) and reduces to it when $k = 1$.

### C.2 DERIVATION OF THE CLOSED-FORM UPDATE

We solve the constrained optimization problem:

$$\min_{h'} \quad \|h' - h\|_2^2 \quad \text{s.t.} \quad U_{\text{prediction}}^\top h' + b_{\text{prediction}} = U_{\text{knowledge}}^\top h + b_{\text{knowledge}}.$$

First, we define the target prediction coordinates

$$p_{\text{target}} := U_{\text{knowledge}}^\top h + b_{\text{knowledge}} - b_{\text{prediction}} \in \mathbb{R}^k,$$

so the constraint becomes

$$U_{\text{prediction}}^\top h' = p_{\text{target}}.$$

Then we form the Lagrangian

$$\mathcal{L}(h', \lambda) = \|h' - h\|_2^2 + \lambda^\top \left( U_{\text{prediction}}^\top h' - p_{\text{target}} \right), \qquad \lambda \in \mathbb{R}^k.$$

Setting the gradient with respect to $h'$ to zero gives

$$2(h' - h) + U_{\text{prediction}} \lambda = 0 \quad \implies \quad h' = h - \tfrac{1}{2} U_{\text{prediction}} \lambda.$$

Substituting into the constraint yields

$$U_{\text{prediction}}^\top \left( h - \tfrac{1}{2} U_{\text{prediction}} \lambda \right) = p_{\text{target}},$$

which is equivalent to

$$U_{\text{prediction}}^\top U_{\text{prediction}} \lambda = 2 \left( U_{\text{prediction}}^\top h - p_{\text{target}} \right).$$

Let

$$G := U_{\text{prediction}}^\top U_{\text{prediction}} \in \mathbb{R}^{k \times k}$$

denote the Gram matrix. When the columns of $U_{\text{prediction}}$ are linearly independent, $G$ is invertible and

$$\lambda = 2G^{-1} \left( U_{\text{prediction}}^\top h - p_{\text{target}} \right).$$

Plugging this into the expression for $h'$ gives the closed-form update

$$h' = h + U_{\text{prediction}} G^{-1} \left( p_{\text{target}} - U_{\text{prediction}}^\top h \right).$$

This expression reduces exactly to the scalar update used in the original binary-choice formulation when $k = 1$.

## D  TRANSFORMED DISTRIBUTION AFTER KAPPA

We visualize how the hidden states are transformed after applying KAPPA by projecting them onto the two-dimensional knowledge-prediction subspace. We consider two representative variants of KAPPA. KAPPA-Linear corresponds to the case with $w > 0$ and $\beta = 0$, where the prediction coordinate is scaled in proportion to the knowledge coordinate. KAPPA-Scalar corresponds to the case with $w = 0$ and $\beta > 0$, where the prediction coordinate is set to a fixed value determined by the sign of the knowledge coordinate. Figure 6 shows the distributions for the original LLM (left), KAPPA-Linear (middle), and KAPPA-Scalar (right).

In the original distribution, correct cases typically appear in the first and third quadrants, where the knowledge and prediction coordinates share the same sign. Incorrect cases, in contrast, appear in the second and fourth quadrants, where the coordinates are misaligned. This observation motivates KAPPA: by shifting hidden states in the direction of the prediction basis, points in the error regions (quadrants II and IV) can be moved toward the correct regions (quadrants I and III), thereby aligning the model's predictions with its internal knowledge and reducing error cases. After applying KAPPA, we indeed observe that the transformed hidden states become more concentrated in the first and third quadrants, demonstrating that knowledge and behavior are effectively aligned through the intervention.

In the KAPPA-Linear case ($w > 0, \beta = 0$), the values on the knowledge and prediction coordinates are aligned along a line, indicating that prediction scores scale proportionally with the knowledge signal. By contrast, in the KAPPA-Scalar case ($w = 0, \beta > 0$), the prediction coordinate is aligned to a fixed value determined by the sign of the knowledge coordinate. This illustrates that KAPPA can induce qualitatively different forms of alignment between knowledge and prediction, depending on the choice of $(w, \beta)$.

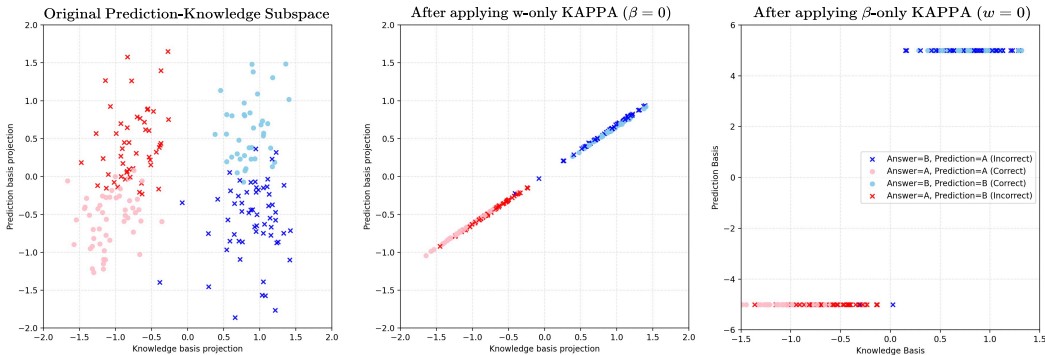

Figure 6: The distribution of hidden states projected to the 2D knowledge-prediction subspace. From the left, LLM, KAPPA-Linear, KAPPA-scalar.

# E    Experimental Setup Details

## E.1    Data Subsets and Counts

We report the train/validation/test splits used in multiple-choice experiments. Table 4 summarizes the number of samples for each dataset. Our dataset maintains perfect class balance by design. Each question is evaluated under both option orders (e.g., "(A) correct, (B) incorrect" and "(A) incorrect, (B) correct"), ensuring that any position bias is neutralized while preserving exact class balance across the entire dataset.

| Dataset | Train | Validation | Test |
|---------|-------|-----------|------|
| BBH | 3,916 | 1,958 | 3,918 |
| MMLU | 9,498 | 4,786 | 9,582 |
| ARC | 2,238 | 598 | 2,344 |

Table 4: Number of samples in the train/validation/test splits for each dataset.

**BBH-Binary Subsets**    We list below all the subsets included in the BBH-Binary benchmark.

| | | |
|---|---|---|
| Boolean Expressions | Date Understanding | Disambiguation QA |
| Geometric Shapes | Hyperbaton | Logical Deduction (5 Objects) |
| Logical Deduction (7 Objects) | Logical Deduction (3 Objects) | Movie Recommendation |
| Navigate | Penguins in a Table | Reasoning about Colored Objects |
| Ruin Names | Salient Translation Error Detection | Sports Understanding |
| Temporal Sequences | Tracking Shuffled Objects (5) | Tracking Shuffled Objects (7) |
| Tracking Shuffled Objects (3) | Web of Lies | |

**MMLU-Binary Subsets**    We list below all the subsets included in the MMLU-Binary benchmark.

| | | |
|---|---|---|
| Abstract Algebra | Anatomy | Astronomy |
| Business Ethics | Clinical Knowledge | College Biology |
| College Chemistry | College Computer Science | College Mathematics |
| College Physics | Computer Security | Conceptual Physics |
| Econometrics | Electrical Engineering | Elementary Mathematics |
| Global Facts | High School Biology | High School Chemistry |
| High School Computer Science | High School Geography | High School Government & Politics |
| High School Macroeconomics | High School Mathematics | High School Physics |
| High School Psychology | High School Statistics | High School US History |
| Human Aging | Human Sexuality | Jurisprudence |
| Management | Marketing | Medical Genetics |
| Miscellaneous | Moral Disputes | Moral Scenarios |
| Nutrition | Philosophy | Prehistory |
| Professional Accounting | Professional Medicine | Professional Psychology |
| Public Relations | Security Studies | Sociology |
| US Foreign Policy | Virology | World Religions |

## E.2    Prompt Formats

To mitigate potential option bias, we varied the prompt format across 32 combinations, derived from four alternative instructions, two answer formats, and four option symbols. We present each component in Table 5, Table 6, and Table 7. To mitigate the effects of option bias—where the model favors a particular choice (Zheng et al., 2024; Pezeshkpour & Hruschka, 2024)—we evaluate the base model across multiple prompt formats (32 combinations), derived from four alternative instructions, two answer formats, and four option symbols. We present each component in Table 5, Table 6, and Table 7. We evaluate all formats and select the format that yields the most balanced predictions on the validation set. All experiments are repeated with permuted answer orders.

## E.3    Prompt Example

We present one example of the prompt format used in our experiments, shown below.

```
You are given a question on college computer science. You are also given
    answer options, out of which only one is correct. You need to answer
    the question by selecting the correct option. You should only answer
    with the choice letter, not the whole answer. Provide your answer
    with (A) or (B), following the output format below.
```

| Instruction | Text |
|---|---|
| 1 | Provide your answer with {option_text}, following the output format below. **Output Format:** {format_text} |
| 2 | Given the following question and candidate answers, choose the correct option. **Output Format:** {format_text} |
| 3 | Choose the correct option between {option_text}. Use the format shown below when you answer. **Output Format:** {format_text} |
| 4 | Choose the best answer. Follow the output format below. **Output Format:** {format_text} |

Table 5: Task instructions used in our experiments.

| Answer Format Type | Format Text |
|---|---|
| 1 | The correct answer is ( |
| 2 | Answer: ( |

Table 6: Answer format types used in experiments.

| Option Symbol Type | Examples |
|---|---|
| Common Alphabet Upper | (A), (B) |
| Roman Numerals | (i), (ii) |
| Common Numbers | (1), (2) |
| Random Alphabet Upper | (Q), (K) |

Table 7: Option symbol types used in experiments.

```
**Output Format**: The correct answer is (A) or The correct answer is (B)

Many cryptographic protocols base their security on assumptions about the
    computational difficulty of integer factorization. Integer
    factorization serves this purpose because we believe that

Choices:
(A): testing primality is computationally intractable
(B): integer multiplication is a function whose inverse, factorization,
    remains difficult for a large class of inputs
```

# F    INTERVENTION EXPERIMENT DETAILS

## F.1    KAPPA CONFIGURATIONS

For our primary evaluation, we select intervention layers based solely on the validation probe accuracy, without consulting the validation performance of the intervened model. This ensures that layer choices are determined independently of downstream results. We consider both single-layer and multi-layer interventions. In the single-layer setting, we intervene at layer 17 for LLaMA-2 and layer 20 for Qwen2.5. For the six-layer setting, we intervene at layers 12–17 for LLaMA-2 and 15–20 for Qwen2.5. This design allows us to test whether interventions at broader depth ranges yield more robust improvements. Regarding hyperparameters, we consider $w$ and $\beta$. For simplicity, we fix $\beta = 0$ (KAPPA-Linear) in all experiments. We set $w = 5.0$ for LLaMA-2 and $w = 8.0$ for Qwen2.5.

## F.2    BASELINE CONFIGURATIONS

**Base.**    The original large language model (LLM) without any intervention or modification. This baseline reflects the model's raw performance on the evaluation datasets.

**Steering.** Activation steering using logistic regression weights, following Li et al. (2023). We train a logistic regression classifier on hidden activations to separate correct vs. incorrect cases, and use the normalized weight vector as a steering direction applied at inference time. The intervention layer is fixed to be identical to the configuration used for KAPPA. The steering strength multiplier is selected from $\{1, 2, 4, 8\}$, tuned on the validation set to achieve the best performance. For generalization experiments, where the probe is trained on one dataset and directly applied to another, we apply the same multiplier value that achieved the best validation performance on the training dataset, without additional tuning on the out-of-distribution dataset.

**LoRA Fine-tuning (LoRA FT).** Parameter-efficient fine-tuning with LoRA adapters applied to the attention projection layers. We fine-tune the model on the same training split as used for our probe training, selecting hyperparameters (rank, learning rate, number of steps) by validation performance. This baseline represents a strong adaptation method requiring training and weight updates. For ARC-Challenge-Binary, we use $r = 16$, $\alpha = 32$, and set LORA dropout to 0,1. We use an effective batch size of 8 in training time, run for 1 epoch with $5e^{-5}$ learning rate. For BBH, we use $r = 8$, $\alpha = 16$, and set LORA dropout to 0,05. We use an effective batch size of 4 in training time, run for 1 epoch with $2e^{-4}$ learning rate. For BBH, we use $r = 64$, $\alpha = 64$, and set LORA dropout to 0,1. We use an effective batch size of 16 in training time, run for 1 epoch with $3e^{-5}$ learning rate.

**In-Context Learning (ICL).** We provide the model with demonstrations retrieved from the training dataset using BM25 retrieval. At inference time, for each test instance, we retrieve $k = 3$ examples and prepend them to the prompt in a few-shot format. This baseline reflects a retrieval-augmented prompting strategy without parameter updates.

## G    AGREEMENT RATE BETWEEN MODEL'S PREDICTIONS AND KNOWLEDGE PROBE

The agreement rate measures how often the model's predicted answer matches the option that the knowledge probe identifies as most supported by the hidden-state representations. Table 8 shows that KAPPA substantially improves this alignment across all binary-choice datasets, indicating that our intervention effectively reduces the knowledge–prediction gap.

| | Llama-2-7B-Chat | | Qwen2.5-7B-Instruct | |
|---|---|---|---|---|
| **Dataset** | Base | KAPPA | Base | KAPPA |
| BBH-Binary | 0.508 | 0.839 | 0.710 | 0.781 |
| MMLU-Binary | 0.641 | 0.800 | 0.808 | 0.885 |
| ARC-Binary | 0.742 | 0.916 | 0.919 | 0.977 |

Table 8: Agreement rate (%) between the model's predicted choice and the option favored by the knowledge probe. Higher values indicate stronger alignment between the model's external predictions and the internal evidence encoded in its hidden states. Results are reported for `Llama-2-7B-Chat` (Layer 17) and `Qwen2.5-7B-Instruct` (Layer 20).

## H    ADDITIONAL PROBING RESULTS

In this section, we provide supplementary probing results to complement the main findings reported in § 4.1. These results further validate that the identified knowledge and prediction directions are consistent across models and datasets. Figure 7, 8 presents results on MMLU-Binary using `Llama-2`, showing that the probe trained to separate correct answers from distractors achieves stable performance across layers. Figure 9, 10 shows results on BBH-Binary for both `Llama-2` (upper) and `Qwen-2.5` (lower). We observe broadly consistent patterns across the two models, indicating that the knowledge-prediction subspace is not specific to a single model family but extends across architectures. Unlike the shared probes used in the main text, here we train a separate probe for each subset individually. While some subsets show clear improvements in probe accuracy, others exhibit limited gains. We attribute these limited improvements to cases where the model itself lacks

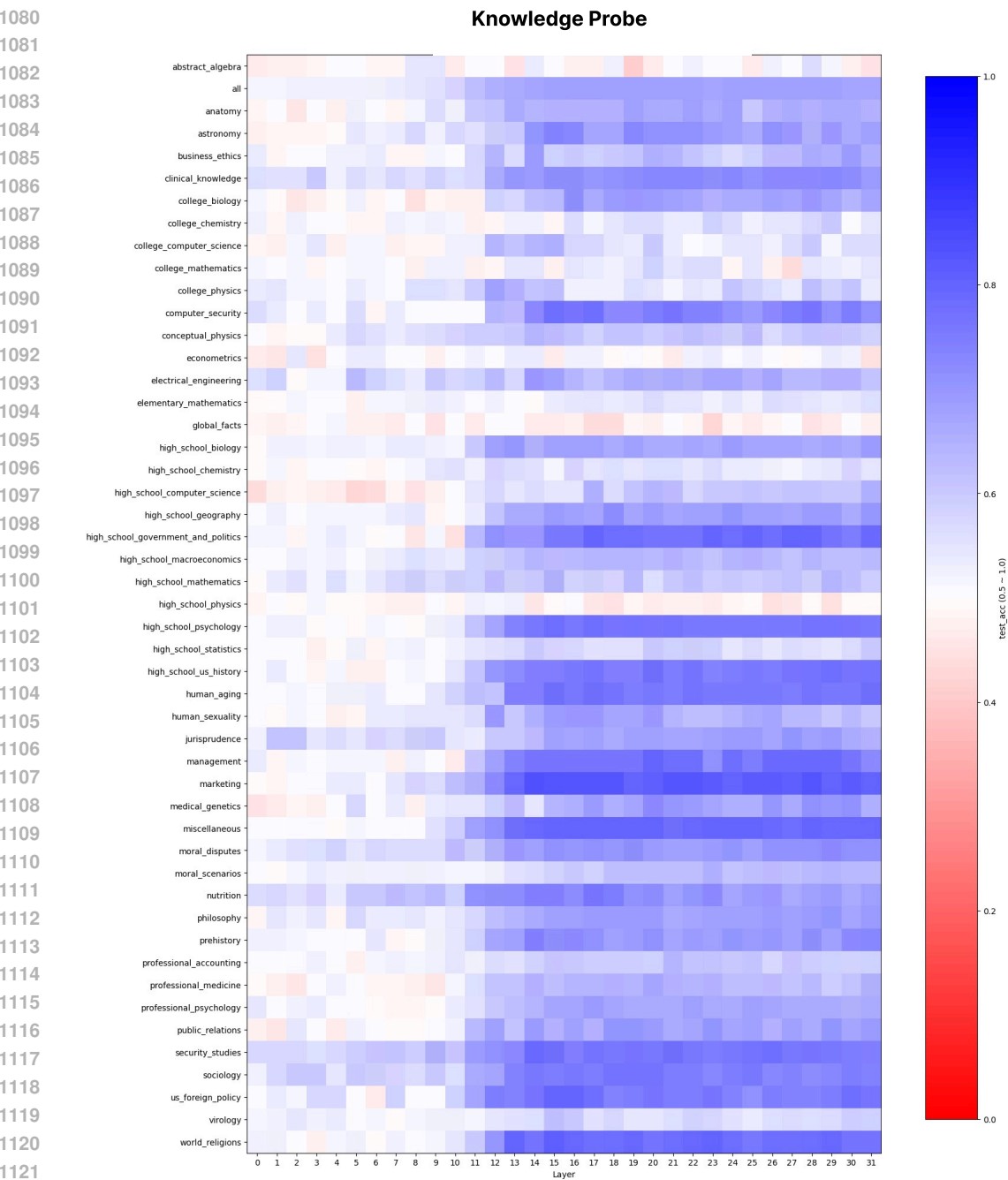

Figure 7: Knowledge probe results on MMLU-Binary using `Llama-2`.

sufficient internal knowledge. Overall, these additional probing results provide further evidence that the knowledge and prediction signals identified by our method are robust, reproducible, and generalizable across datasets and model backbones.

## I    ADDITIONAL INTERVENTION RESULTS

We further report additional results in Figure 11 using the `Llama-2` model on the BBH-Binary dataset. In this setting, we train a separate probe for each subset and validate across eight layer configurations (12, 13, 14, 15, 16, 17, 15–17, and 12–17), selecting the best-performing layer per

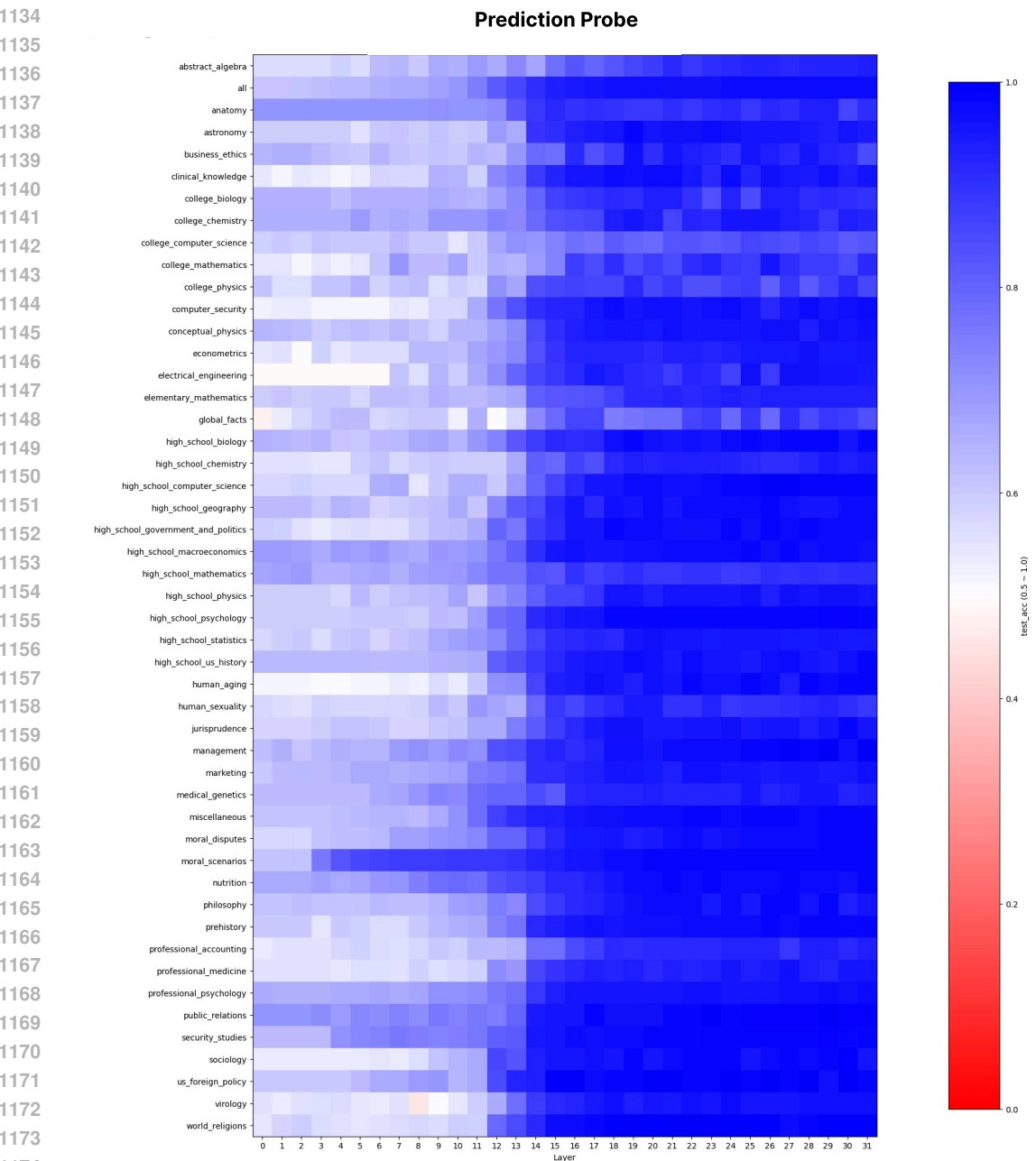

Figure 8: Prediction probe results on MMLU-Binary using `Llama-2`.

subset. Applying KAPPA-Scalar ($w = 0, \beta = 5$), we achieve 75.3% accuracy, which even surpass the accuracy of the best knowledge probe trained jointly on all subsets (72.8%). This indicates that while a shared subspace exists across different reasoning tasks, the optimal subspace varies by task, suggesting that there remains room for further optimization when applying our method to downstream tasks. We further observe that knowledge probe accuracy varies across subsets, and the intervened accuracy tends to increase accordingly, following the same trend. This indicates that our method effectively aligns model behavior with the knowledge captured by the probe.

## J    FULL INTERVENTION RESULTS

Detailed results for all datasets are provided in Table 9 (`Llama-2`) and Table 10 (`Qwen-2.5`).

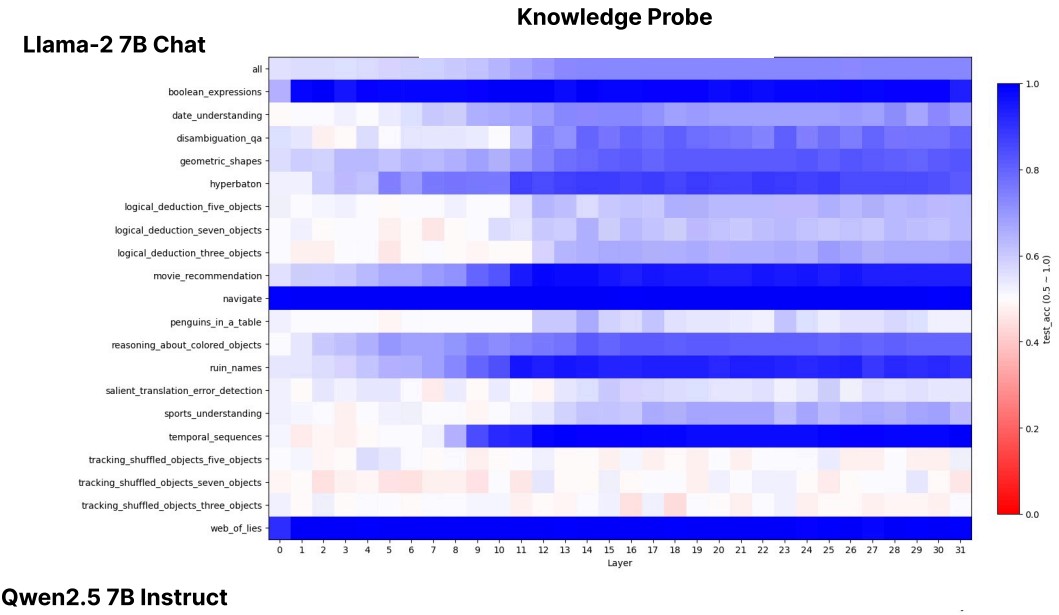

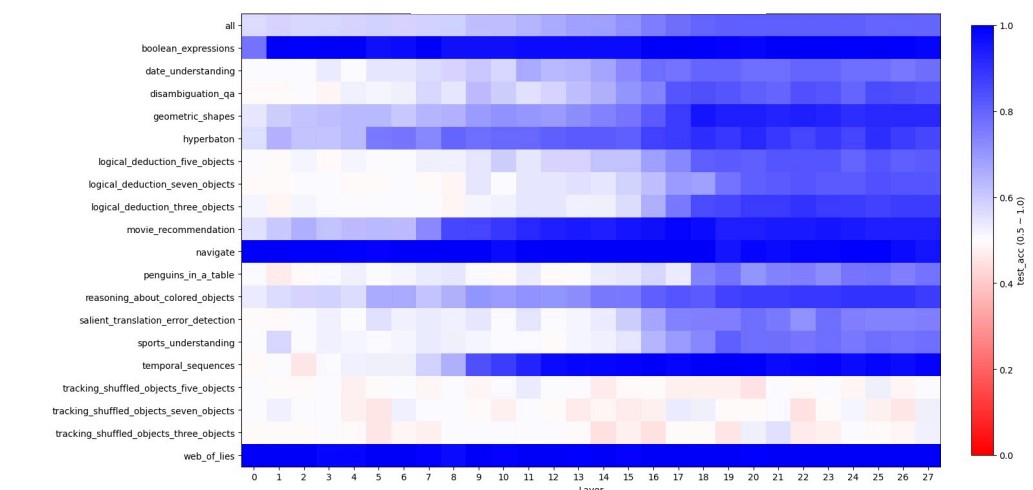

Figure 9: Knowledge probe results on BBH-Binary using `Llama-2` (upper) and `Qwen-2.5` (lower).

Table 9: `Llama-2` full intervention results. Accuracy (%) across datasets with different intervention layers.

| Dataset - Subset | Base Acc | 1-Layer (17) | 6-Layers (12–17) |
|---|---|---|---|
| ARC - Challenge | 74.1 | 76.3 | 75.8 |
| BBH - Boolean Expressions | 45.0 | 92.5 | 96.5 |
| BBH - Date Understanding | 66.0 | 66.5 | 65.5 |
| BBH - Disambiguation QA | 67.0 | 73.0 | 71.5 |
| BBH - Geometric Shapes | 69.5 | 57.0 | 68.5 |
| BBH - Hyperbaton | 61.0 | 78.5 | 77.0 |
| BBH - Logical Deduction (5 Objects) | 55.0 | 61.0 | 60.0 |
| BBH - Logical Deduction (7 Objects) | 59.5 | 60.0 | 62.5 |
| BBH - Logical Deduction (3 Objects) | 59.0 | 64.0 | 63.5 |
| BBH - Movie Recommendation | 74.0 | 82.0 | 87.0 |
| BBH - Navigate | 4.5 | 99.0 | 98.0 |

*Continued on next page*

Table 9: (continued)

| Dataset - Subset | Base Acc | 1-Layer (17) | 6-Layers (12–17) |
|---|---|---|---|
| BBH - Penguins in a Table | 54.2 | 58.5 | 60.2 |
| BBH - Reasoning About Colored Objects | 81.0 | 79.0 | 78.0 |
| BBH - Ruin Names | 41.0 | 76.5 | 82.0 |
| BBH - Salient Translation Error Detection | 48.5 | 55.5 | 55.5 |
| BBH - Sports Understanding | 61.0 | 53.5 | 57.0 |
| BBH - Temporal Sequences | 21.0 | 95.5 | 91.5 |
| BBH - Tracking Shuffled Objects (5 Objects) | 49.5 | 51.0 | 49.0 |
| BBH - Tracking Shuffled Objects (7 Objects) | 50.5 | 49.0 | 50.0 |
| BBH - Tracking Shuffled Objects (3 Objects) | 45.5 | 50.5 | 50.5 |
| BBH - Web of Lies | 5.0 | 93.0 | 91.5 |
| **BBH - Average** | 50.9 | 69.8 | 70.8 |
| MMLU - Abstract Algebra | 63.8 | 52.5 | 47.5 |
| MMLU - Anatomy | 63.0 | 63.9 | 64.8 |
| MMLU - Astronomy | 68.0 | 68.0 | 68.9 |
| MMLU - Business Ethics | 66.3 | 60.0 | 65.0 |
| MMLU - Clinical Knowledge | 72.2 | 75.5 | 74.1 |
| MMLU - College Biology | 66.4 | 69.0 | 65.5 |
| MMLU - College Chemistry | 60.0 | 55.0 | 50.0 |
| MMLU - College Computer Science | 57.5 | 63.8 | 65.0 |
| MMLU - College Mathematics | 50.0 | 53.8 | 52.5 |
| MMLU - College Physics | 42.7 | 46.3 | 51.2 |
| MMLU - Computer Security | 72.5 | 71.3 | 75.0 |
| MMLU - Conceptual Physics | 60.6 | 61.7 | 61.7 |
| MMLU - Econometrics | 54.3 | 53.3 | 54.4 |
| MMLU - Electrical Engineering | 52.6 | 59.5 | 61.2 |
| MMLU - Elementary Mathematics | 53.0 | 54.0 | 54.6 |
| MMLU - Global Facts | 55.0 | 55.0 | 55.0 |
| MMLU - High School Biology | 68.5 | 67.7 | 69.0 |
| MMLU - High School Chemistry | 54.3 | 56.1 | 56.7 |
| MMLU - High School Computer Science | 60.0 | 60.0 | 61.3 |
| MMLU - High School Geography | 68.8 | 68.8 | 70.6 |
| MMLU - High School Government and Politics | 77.6 | 77.6 | 80.1 |
| MMLU - High School Macroeconomics | 60.6 | 62.5 | 67.0 |
| MMLU - High School Mathematics | 48.6 | 52.8 | 55.6 |
| MMLU - High School Physics | 50.0 | 49.2 | 49.2 |
| MMLU - High School Psychology | 75.2 | 76.2 | 77.3 |
| MMLU - High School Statistics | 55.2 | 58.6 | 58.1 |
| MMLU - High School US History | 73.2 | 74.4 | 74.4 |
| MMLU - Human Aging | 78.3 | 75.0 | 75.6 |
| MMLU - Human Sexuality | 60.4 | 68.9 | 71.7 |
| MMLU - Jurisprudence | 64.8 | 67.1 | 67.1 |
| MMLU - Management | 69.0 | 75.0 | 77.4 |
| MMLU - Marketing | 75.0 | 78.2 | 84.0 |
| MMLU - Medical Genetics | 60.0 | 65.0 | 65.0 |
| MMLU - Miscellaneous | 76.4 | 77.9 | 78.0 |
| MMLU - Moral Disputes | 70.5 | 74.5 | 73.0 |
| MMLU - Moral Scenarios | 49.2 | 49.4 | 50.6 |
| MMLU - Nutrition | 62.2 | 64.2 | 69.9 |
| MMLU - Philosophy | 65.2 | 67.2 | 68.8 |
| MMLU - Prehistory | 66.9 | 70.0 | 72.7 |
| MMLU - Professional Accounting | 59.7 | 58.4 | 59.7 |
| MMLU - Professional Medicine | 66.1 | 67.4 | 67.4 |
| MMLU - Professional Psychology | 62.9 | 67.1 | 66.9 |
| MMLU - Public Relations | 68.2 | 70.5 | 72.7 |
| MMLU - Security Studies | 62.8 | 65.8 | 68.9 |
| MMLU - Sociology | 75.3 | 75.3 | 75.9 |
| MMLU - US Foreign Policy | 72.5 | 75.0 | 77.5 |
| MMLU - Virology | 60.4 | 59.7 | 64.9 |
| MMLU - World Religions | 79.0 | 78.3 | 82.6 |

*Continued on next page*

Table 9: (continued)

| Dataset - Subset | Base Acc | 1-Layer (17) | 6-Layers (12–17) |
|---|---|---|---|
| **MMLU - Average** | 63.6 | 64.9 | 66.2 |

Table 10: `Qwen2.5` full intervention results. Accuracy (%) across datasets with different intervention layers.

| Dataset - Subset | Base Acc | 1-Layer (20) | 6-Layers (15-20) |
|---|---|---|---|
| ARC - Challenge | 91.9 | 93.1 | 93.5 |
| BBH - Boolean Expressions | 51.0 | 51.5 | 51.5 |
| BBH - Date Understanding | 76.5 | 76.5 | 76.5 |
| BBH - Disambiguation QA | 74.5 | 74.0 | 74.5 |
| BBH - Geometric Shapes | 77.5 | 79.5 | 81.5 |
| BBH - Hyperbaton | 64.5 | 68.0 | 75.0 |
| BBH - Logical Deduction (5 Objects) | 79.0 | 81.5 | 83.0 |
| BBH - Logical Deduction (7 Objects) | 76.5 | 80.0 | 81.5 |
| BBH - Logical Deduction (3 Objects) | 83.5 | 83.0 | 84.0 |
| BBH - Movie Recommendation | 57.5 | 59.5 | 71.0 |
| BBH - Navigate | 90.5 | 91.0 | 92.5 |
| BBH - Penguins in a Table | 74.6 | 72.9 | 73.7 |
| BBH - Reasoning About Colored Objects | 86.0 | 87.0 | 89.0 |
| BBH - Ruin Names | 61.5 | 63.5 | 69.0 |
| BBH - Salient Translation Error Detection | 71.5 | 72.5 | 72.0 |
| BBH - Sports Understanding | 73.5 | 73.5 | 73.5 |
| BBH - Temporal Sequences | 80.0 | 85.0 | 93.0 |
| BBH - Tracking Shuffled Objects (5 Objects) | 49.0 | 51.0 | 50.0 |
| BBH - Tracking Shuffled Objects (7 Objects) | 51.0 | 51.0 | 49.5 |
| BBH - Tracking Shuffled Objects (3 Objects) | 45.5 | 45.5 | 44.5 |
| BBH - Web of Lies | 98.5 | 100.0 | 100.0 |
| **BBH - Average** | 71.1 | 72.3 | 74.3 |
| MMLU - Abstract Algebra | 65.0 | 66.3 | 60.0 |
| MMLU - Anatomy | 76.9 | 77.8 | 78.7 |
| MMLU - Astronomy | 90.2 | 93.4 | 93.4 |
| MMLU - Business Ethics | 81.3 | 78.8 | 77.5 |
| MMLU - Clinical Knowledge | 92.5 | 92.9 | 92.0 |
| MMLU - College Biology | 96.6 | 94.8 | 94.8 |
| MMLU - College Chemistry | 67.5 | 63.8 | 67.5 |
| MMLU - College Computer Science | 82.5 | 80.0 | 83.8 |
| MMLU - College Mathematics | 62.5 | 73.8 | 72.5 |
| MMLU - College Physics | 69.5 | 75.6 | 73.2 |
| MMLU - Computer Security | 81.3 | 82.5 | 83.8 |
| MMLU - Conceptual Physics | 81.9 | 85.1 | 84.0 |
| MMLU - Econometrics | 78.3 | 83.7 | 84.8 |
| MMLU - Electrical Engineering | 77.6 | 79.3 | 80.2 |
| MMLU - Elementary Mathematics | 80.9 | 80.9 | 82.2 |
| MMLU - Global Facts | 62.5 | 62.5 | 60.0 |
| MMLU - High School Biology | 89.5 | 90.3 | 90.7 |
| MMLU - High School Chemistry | 78.7 | 75.6 | 79.9 |
| MMLU - High School Computer Science | 78.8 | 80.0 | 80.0 |
| MMLU - High School Geography | 86.9 | 89.4 | 90.6 |
| MMLU - High School Gov. & Politics | 95.5 | 95.5 | 93.6 |
| MMLU - High School Macroeconomics | 88.8 | 89.0 | 88.8 |
| MMLU - High School Mathematics | 66.2 | 68.5 | 69.0 |
| MMLU - High School Physics | 78.7 | 78.7 | 77.9 |
| MMLU - High School Psychology | 89.5 | 91.3 | 92.7 |
| MMLU - High School Statistics | 82.2 | 86.2 | 86.8 |
| MMLU - High School US History | 90.2 | 91.5 | 92.1 |
| MMLU - Human Aging | 83.3 | 83.3 | 85.0 |
| MMLU - Human Sexuality | 85.9 | 89.6 | 89.6 |

*Continued on next page*

Table 10: (continued)

| Dataset - Subset | Base Acc | 1-Layer (20) | 6-Layers (15-20) |
|---|---|---|---|
| MMLU - Jurisprudence | 81.8 | 84.1 | 85.2 |
| MMLU - Management | 86.9 | 92.9 | 94.1 |
| MMLU - Marketing | 93.1 | 93.1 | 93.6 |
| MMLU - Medical Genetics | 90.0 | 88.8 | 90.0 |
| MMLU - Miscellaneous | 88.2 | 89.7 | 90.1 |
| MMLU - Moral Disputes | 80.9 | 85.3 | 84.2 |
| MMLU - Moral Scenarios | 53.6 | 55.2 | 54.2 |
| MMLU - Nutrition | 82.9 | 85.8 | 86.6 |
| MMLU - Philosophy | 80.0 | 83.2 | 81.2 |
| MMLU - Prehistory | 84.6 | 86.5 | 86.9 |
| MMLU - Professional Accounting | 73.5 | 72.6 | 73.9 |
| MMLU - Professional Medicine | 83.9 | 83.5 | 85.3 |
| MMLU - Professional Psychology | 80.8 | 83.5 | 83.9 |
| MMLU - Public Relations | 83.0 | 88.6 | 89.8 |
| MMLU - Security Studies | 76.0 | 84.7 | 84.2 |
| MMLU - Sociology | 90.1 | 89.5 | 88.3 |
| MMLU - US Foreign Policy | 88.8 | 91.3 | 95.0 |
| MMLU - Virology | 68.7 | 69.4 | 70.2 |
| MMLU - World Religions | 92.8 | 94.9 | 97.1 |
| **MMLU - Average** | 81.3 | 83.0 | 83.3 |

## K  STEERING BASELINE EXPERIMENTAL DETAILS

We summarize the steering baseline configuration in Table 11. For both `Llama-2` and `Qwen2.5`, we fix the intervention layer in advance, normalize each probe weight vector, and choose the best-performing multiplier from $1, 2, 4, 8$ using validation accuracy on the training dataset. When applying to other datasets for generalization, we consistently use the same layer and multiplier configuration without additional tuning.

Table 11: Configuration of the Steering baseline. We fix the intervention layer per model, normalize the probe vector, and select the best multiplier from $\{1, 2, 4, 8\}$ based on validation performance. The same configuration is applied when generalizing to other datasets.

| Model | Training Dataset | Layer | Multiplier |
|---|---|---|---|
| `Llama-2` | BBH | 17 | 8.0 |
| | MMLU | 17 | 4.0 |
| | ARC | 17 | 1.0 |
| `Qwen2.5` | BBH | 20 | 1.0 |
| | MMLU | 20 | 4.0 |
| | ARC | 20 | 4.0 |

## L  MULTIPLE-CHOICE INTERVENTION EXPERIMENT

### L.1  EXPERIMENTAL SETUP DETAILS

We provide additional details on the setup of our multiple-choice intervention experiments presented in § 4.3.

**Dataset Details.** We evaluate on the same MMLU and ARC-Challenge subsets described in Appendix E, using their original multiple-choice configurations (4 options for MMLU and 3 options for ARC-Challenge). For the BBH benchmark, we exclude subsets containing more than three answer options, resulting in the following ten subsets:

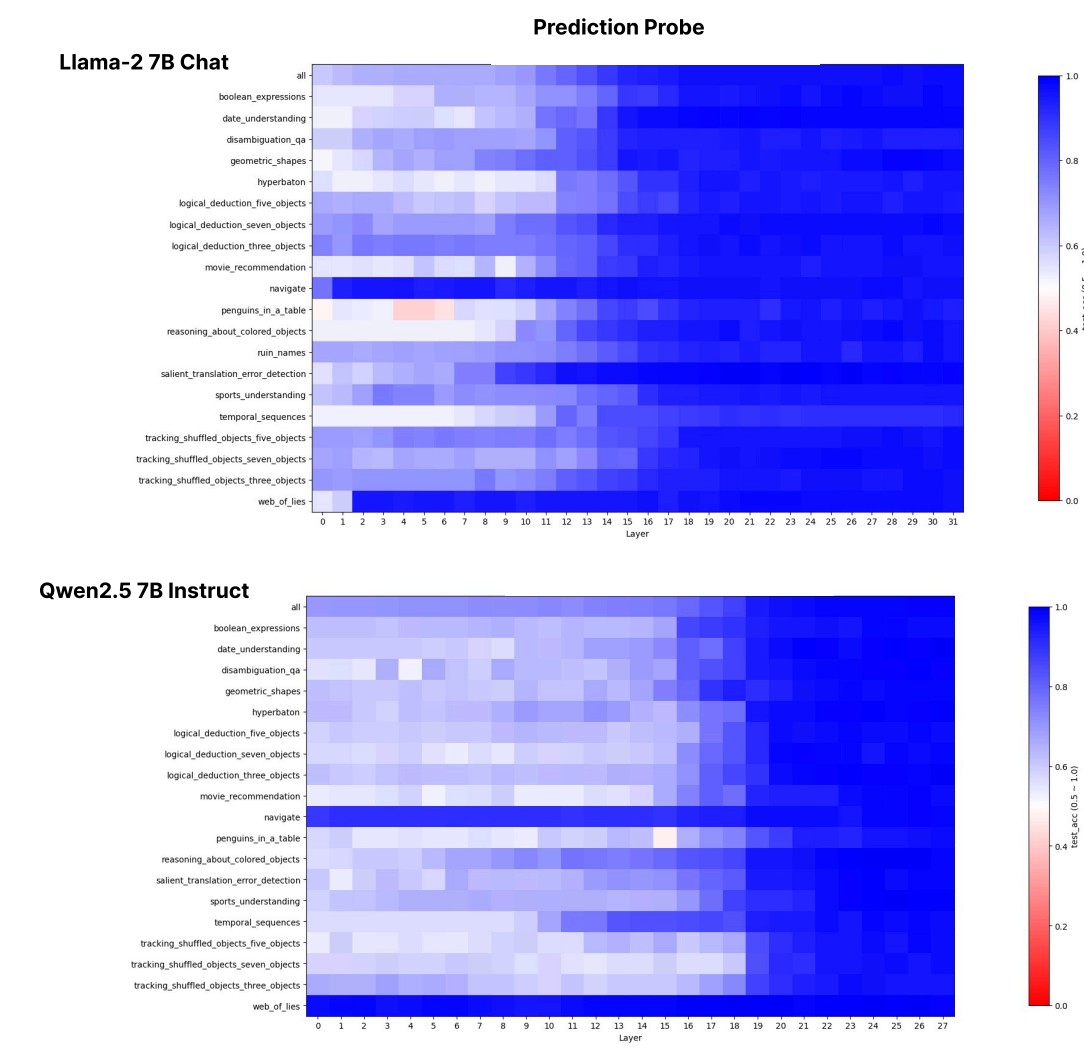

Figure 10: Prediction probe results on BBH-Binary using `Llama-2` (upper) and `Qwen-2.5` (lower).

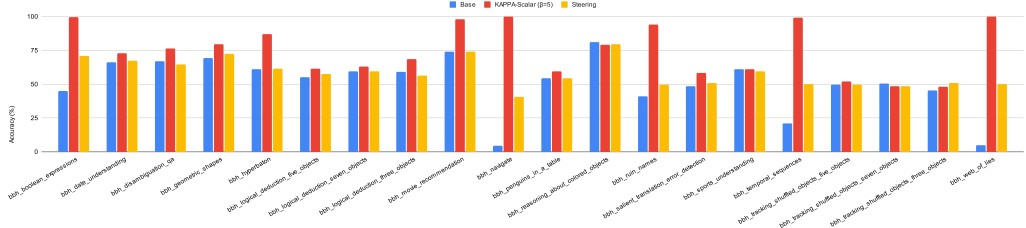

Figure 11: Subset-wise results on the BBH-Binary dataset with `Llama-2`. For each subset, we train a separate knowledge probe and apply KAPPA-Scalar ($w = 0, \beta = 5$) at the best-performing layer. We observe that higher knowledge probe accuracy is accompanied by higher intervened accuracy, indicating that our method effectively aligns the model's predictions with the knowledge captured by the probe.

| Date Understanding | Geometric Shapes | Logical Deduction (5 Objects) |
| Logical Deduction (7 Objects) | Movie Recommendation | Penguins in a Table |
| Salient Translation Error Detection | Temporal Sequences | Tracking Shuffled Objects (5) |
| Tracking Shuffled Objects (7) | | |

We use the same prompt format as in the binary-choice experiments (Appendix E.2) and evaluate each MCQ item under eight randomly sampled permutation orders. Table 12 summarizes the size of each dataset split.

| Dataset | Train | Validation | Test |
|---------|-------|------------|------|
| BBH | 7,664 | 3,832 | 7,672 |
| MMLU | 35,592 | 17,920 | 35,872 |
| ARC | 6,714 | 1,794 | 7,032 |

Table 12: Number of samples in the train/validation/test splits for each dataset used for multiple-choice intervention.

**Hyperparameter Details.** The intervention layers were selected as the contiguous layer range that achieved the highest knowledge-probe accuracy on the validation split. We set the hyperparameters uniformly across all experiments, using $\beta = 0$, $w = 30$ for the single-layer case, and $w = 20$ for the multi-layer (3-layer) setting. The corresponding probe learning rates are also reported in Table 13.

| Model & Dataset | 1-layer | 3-layer | Learning Rate |
|-----------------|---------|---------|---------------|
| Llama–BBH | 15 | 15–17 | $5e^{-4}$ |
| Llama–MMLU | 17 | 17–19 | $2e^{-4}$ |
| Llama–ARC | 18 | 17–19 | $2e^{-4}$ |
| Qwen–BBH | 21 | 20–22 | $5e^{-4}$ |
| Qwen–MMLU | 21 | 21–23 | $2e^{-4}$ |
| Qwen–ARC | 22 | 22–24 | $2e^{-4}$ |

Table 13: Hyperparameter settings for layer selection and probe learning rates across models and datasets used in multiple-choice intervention experiments.

## M  FREE-FORM GENERATION DETAILS

### M.1  EXPERIMENTAL DETAILS

**Dataset Construction.** We construct a free-form version of the BBH dataset by systematically converting MCQ items into open-ended questions. To ensure valid evaluation, we remove questions whose semantics inherently require choosing among options (e.g., "Which statement best explains..."). This yields a total of 1,759 valid free-form reasoning questions, providing a rigorous testbed for assessing generative reasoning without multiple-choice cues. In this setting, KAPPA operates throughout decoding: at each generation step, the hidden state is projected onto the knowledge–prediction subspace, the mismatch between the two axes is detected, and the representation is dynamically adjusted. Thus, KAPPA influences the entire generation trajectory rather than a single output decision.

**Evaluation Methodology.** For assessment, we employ an LLM-as-a-Judge approach, following established practices in recent work (Chandak et al., 2025). The judge model evaluates whether the ground-truth answer is semantically contained within or equivalent to the model's generated response, allowing for flexible matching that accounts for varied expression while maintaining accuracy standards.

### M.2  EVALUATION PROMPT

```
Your task is to judge whether the given response to a question matches a
    given ground truth answer or not. You are provided with a question, a
     ground truth response, and the response you need to judge.
For a response to "match", it must have at least as much information as
    the ground-truth.
The response can have more information than the ground-truth. It can be
    more specific (for example, "Labrador" is more specific than "dog"),
```

```
or have additional possible correct answers. But it must cover
    everything mentioned in the ground-truth. It is okay if it covers it
    in different words, i.e. paraphrased.
For numeric answers, the relative error, defined as |response - ground
    truth| / mean(response, ground truth), must be less than 1\% for the
    response to be judged as a correct match. Here, if the ground truth
    is a specific numeric quantity but the response is a range, then they
     don't match (even if the range contains the ground truth).

Possible judgments:

"0": The response does not match the ground-truth answer.
"1": The response matches the ground-truth.

Question: [question]
Ground truth: [target]
Response: [response]

Your job is to ONLY check whether the given response matches the ground
    truth answer or not in the context of the question. You DO NOT NEED
    to assess the correctness of the response. This is part of an
    automated evaluation process, therefore you MUST OUTPUT your final
    answer as "0" or "1" in <answer> </answer> tags.
```

## N  GENERAL CAPABILITY EVALUATION

### N.1  DATASET DETAILS

The `vicuna-eval` (Chiang et al., 2023) benchmark consists of 80 unique questions spanning 9 categories. To ensure robustness and remove position bias, we adopt a two-fold evaluation protocol: each question is evaluated twice with swapped response orders, yielding 160 total comparisons. Each comparison is conducted as a first-turn message in a fresh session. This evaluation protocol ensures fairness by controlling for both position bias and category imbalance. The overall win rate (46.9% for KAPPA vs. 45.2% for LoRA) is computed over all 160 comparisons.

| Category | Count |
|---|---|
| generic | 20 |
| knowledge | 20 |
| roleplay | 20 |
| common-sense | 20 |
| fermi | 20 |
| counterfactual | 20 |
| coding | 14 |
| math | 6 |
| writing | 20 |
| **Total** | **160** |

Table 14: Distribution of question categories.

### N.2  EVALUATION PROMPT

We include the full prompts used for evaluation across different categories (*general*, *coding*, and *math*). For each category, we present three components: (i) the **system prompt**, (ii) the **format prompt**, which specifies the evaluation structure, and (iii) the **task instruction**, describing the specific evaluation criteria. In constructing the final evaluation input, the format prompt is concatenated with the task instruction to form the **user message**. Within the format prompt, the original question is inserted together with two model responses: one from KAPPA and one from the baseline. To mitigate order bias, both possible permutations of the two responses are provided to the evaluator.

**System Prompt:**

```
1566    You are a helpful and precise assistant for checking the quality of the
1567        answer.
1568
1569    Format:
1570
1571    [Question]
1572    {question}
1573
1574    [The Start of Assistant 1's Answer]
1575    {answer_1}
        [The End of Assistant 1's Answer]
1576
1577    [The Start of Assistant 2's Answer]
1578    {answer_2}
        [The End of Assistant 2's Answer]
1579
1580    Generic Task Instructions:
1581
1582    We would like to request your feedback on the performance of two AI
1583        assistants
        in response to the user question displayed above.
1584    Please rate the helpfulness, relevance, accuracy, level of details of
1585        their responses.
1586    Each assistant receives an overall score on a scale of 1 to 10, where a
1587        higher score
1588    indicates better overall performance.
        Please first output a single line containing only two values indicating
1589        the scores
1590    for Assistant 1 and 2, respectively. The two scores are separated by a
1591        space.
1592    In the subsequent line, please provide a comprehensive explanation of
1593        your evaluation,
        avoiding any potential bias and ensuring that the order in which the
1594        responses were
1595    presented does not affect your judgment.
1596
1597    Coding Task Instructions:
1598
1599    Your task is to evaluate the coding abilities of the above two assistants
1600        . They have been asked to implement a program to solve a given
        problem. Please review their code submissions, paying close attention
1601         to their problem-solving approach, code structure, readability, and
1602        the inclusion of helpful comments.
1603
1604    Please ensure that the assistants' submissions:
        - Correctly implement the given problem statement.
1605    - Contain accurate and efficient code.
1606    - Include clear and concise comments that explain the code's logic and
1607        functionality.
1608    - Adhere to proper coding standards and best practices.
1609
1610    Once you have carefully reviewed both submissions, provide detailed
        feedback on their strengths and weaknesses, along with any
1611        suggestions for improvement. You should first output a single line
1612        containing two scores on the scale of 1-10 (1: no code/no sense; 10:
1613        perfect) for Assistant 1 and 2, respectively. Then give extra
1614        comments starting from the next line.
1615
1616    Mathematical Task Instructions:
1617
1618    We would like to request your feedback on the mathematical proficiency of
         two AI assistants regarding the given user question displayed above.
1619    First, please solve the problem independently, without referring to the
         answers provided by Assistant 1 and Assistant 2.
```

```
Afterward, please examine the problem-solving process of Assistant 1 and
    Assistant 2 step-by-step to ensure their correctness, identifying any
     incorrect steps if present. Your evaluation should take into account
      not only the answer but also the problem-solving steps.
Finally, please output a Python tuple containing two numerical scores for
    Assistant 1 and Assistant 2, ranging from 1 to 10, respectively. If
    applicable, explain the reasons for any variations in their scores
    and determine which assistant performed better.
```

### N.3  RESULTS

Results on general capability evaluations are presented in Table 15. KAPPA consistently achieves win rates above 50% in subsets which require models to retrieve facts and perform structured reasoning—whether through quantitative estimation, factual recall, or counterfactual inference. This highlights its strength in faithfully leveraging internal knowledge during generation.

| Category | vs Steering (%) | vs LoRA FT (%) |
|---|---|---|
| Overall | **51.2** | 46.9 |
| fermi | **65.0** | **65.0** |
| counterfactual | **60.0** | **75.0** |
| knowledge | **55.0** | **55.0** |
| coding | **57.1** | 28.6 |
| math | **50.0** | 33.3 |
| generic | **60.0** | 45.0 |
| common-sense | 40.0 | 25.0 |
| roleplay | 40.0 | 40.0 |
| writing | 35.0 | 40.0 |

Table 15: Win rates of KAPPA against Steering and LoRA FT across categories in `vicuna-eval`. Bold numbers indicate categories where KAPPA achieves at least 50% wins.

## O  RESULTS ACROSS A BROADER SET OF MODELS

To evaluate whether our findings generalize across scales and newer architectures, we additionally test `Qwen3-32B`, `Llama-3.1-8B`, and `Qwen3-4B` on BBH-Binary. Across all model families, we consistently observe substantial knowledge–prediction gaps, and KAPPA reliably narrows these gaps, bringing model predictions significantly closer to the knowledge probe. We follow the same experimental settings as in our other binary-choice evaluations, using a hyperparameter of $w = 20$.

| Method | Qwen3-32B | Llama-3.1-8B | Llama-2-7B | Qwen2.5-7B | Qwen3-4B |
|---|---|---|---|---|---|
| Base | 75.3 | 68.0 | 50.9 | 71.1 | 64.2 |
| 1 Layer | 76.4 | 80.7 | 69.8 | 72.3 | 76.2 |
| 6 Layer | 84.4 | 81.3 | 70.8 | 74.3 | 77.9 |
| Knowledge Probe | 85.5 | 81.5 | 72.8 | 81.0 | 79.6 |

Table 16: Test accuracies across multiple models on the BBH Binary benchmark, using the Instruct/Chat variants for all models. Columns for `Llama-2-7B` and `Qwen2.5-7B` are taken directly from the original paper.

