# OpenReview forum: "Bridging the Knowledge-Prediction Gap in LLMs on Multiple-Choice Questions"
_ICLR.cc/2026/Conference — Submitted to ICLR 2026_

### Official Review · Reviewer_2gxg · 2025-10-16

**Soundness:** 3
**Presentation:** 3
**Contribution:** 1
**Rating:** 2
**Confidence:** 4

**Summary:**

This work proposes a method for intervening on LLM inference by identifying separate basis vectors for an LLM's internal knowledge, and its task prediction. The work suggests that a major source of error during reasoning is not in the knowledge domain but the prediction. The LLM knows the necessary information, but somehow still does not make the correct prediction. The method aims to align these vectors so that prediction is more representative of its knowledge.

**Strengths:**

The paper is well laid out. The motivation behind the method in general, as well as many of its specific design choices, is clear. Experimental results are understandable and reasonably in-depth, with the exception of the free-form results.

**Weaknesses:**

1. Experimental results demonstrate that the method consistently underperforms several alternatives in the multiple-choice setting.
2. Some unsupported claims are made during the analysis of the main-table results. l. 308: "these probes serve approximate upper bounds for each model’s achievable performance without any external information". This phrase makes several key claims which seem problematic.
(a) "approximate upper bounds": the probe is clearly not the upper bound, since fine-tuning outperforms (b) "without external information": presumably this condition is provided to exclude fine-tuning from the claim, but I'm not sure this makes sense. "External information" is not defined, and its not clear how fine-tuning introduces more information than training probes does: they both leverage information from their training set.
3. While results look better for free-form (since the probe is not applicable anymore), the results presented here are unclear. Which dataset is being tested on? Wouldn't it make more sense to test long-form QA tasks (e.g. summarization) in order to evaluate free-form response?
4. Justification for binarization of the datasets is never provided.

**Questions:**

1. l.393 "KAPPA’s success is not merely an artifact of manipulating binary choices but stems from an enhancement of the model’s ability to express its knowledge, indicating its potential to improve performance across a broader range of generative tasks". What do you mean by "manipulating binary choices"? This claim doesn't really connect with your previous argumentation that the fact that models will correctly answer in free-form and then mis-classify in a MCQA format, indicating prediction failure.

---

> ### Author Response · Authors · 2025-12-03
> **Official Response to Reviewer (Part 1of 5)**
>
> We thank the reviewer for the time spent reviewing our work. We appreciate the recognition that our writing is well laid out, our motivation is general, our design choices are clear, and our experimental results are in-depth. We are sincerely grateful that our efforts have been acknowledged.
>
> ---
>
> > **W1.** Experimental results demonstrate that the method consistently underperforms several alternatives in the multiple-choice setting.
>
> **A1.** We appreciate the reviewer's careful examination of our results. We would like to clarify that the primary goal of KAPPA is not to maximize task accuracy, **but to diagnose and bridge the "knowledge-prediction gap" through geometric alignment.** Hence, it is expected that our method underperforms compared to fine-tuning the model, which injects new knowledge to the model. We included fine-tuning only as a point of reference (more like an upper bound), not as a directly comparable baseline.
>
> When evaluated against our objective, our method demonstrates its effectiveness through the following evidence: (i) KAPPA consistently aligns the model’s predictions with the knowledge probe, (ii) it reduces cases where the model answers correctly in free-form but fails in MCQ, and (iii) it achieves these gains while using far fewer parameters than LoRA. More details are as follows.
>
> **(1) Evidence of Efficacy I: Reaching the Intrinsic Knowledge Limit**
>
> Our method is designed to ensure that the knowledge encoded along the knowledge basis is faithfully reflected in the model's actual predictions. Thus, the natural performance target is the knowledge probe's accuracy, which represents what the model internally knows based on its hidden representation rather than state-of-the-art task accuracy.
>
> Empirically, KAPPA makes the model’s answer choices substantially more consistent with the probe’s predictions. This improvement is quantitatively captured by the agreement rate between the knowledge probe’s predicted answers and the model’s actual outputs.
>
> \
> *Table 1: Agreement Rate (%) on Llama-2 7B Chat (Layer 17) and Qwen2.5 7B Instruct (Layer 20)).*
> | Dataset      | Llama-2 Base | Llama-2 KAPPA | Qwen-2.5 Base | Qwen-2.5 KAPPA |
> |--------------|--------------|----------------|----------------|-----------------|
> | BBH-Binary   | 0.508        | 0.839          | 0.710          | 0.781           |
> | MMLU-Binary  | 0.641        | 0.800          | 0.808          | 0.885           |
> | ARC-Binary   | 0.742        | 0.916          | 0.919          | 0.977           |
>
> \
> These result confirms two key findings:
> 1. The prediction basis has a causal effect on the model's behavior
> 2. Modifying the hidden representation only within this 2D subspace is sufficient to make the model's predictions reflect its latent knowledge
>
> This demonstrates that KAPPA successfully bridges the gap between what the model knows and what it predicts, fulfilling its design objective.
>
> \
> **(2) Evidence of Efficacy II: Precision in Mitigating the Specific Gap**
>
> Furthermore, KAPPA is more precise than fine-tuning in addressing the specific knowledge-prediction gap. As shown in Figure 4a, while LoRA reduces the "free-form correct → MCQ incorrect" gap rate from 11.4% to 5.7%, KAPPA reduces it further to 5.1%.
>
> This indicates that while LoRA improves general capacity through parameter updates, KAPPA more directly targets and resolves the structural misalignment we aim to address—cases where the model possesses correct knowledge but fails to express it faithfully in its predictions.
>
> \
> **(3) Validation of Hypothesis: Efficiency as Proof**
>
> Notably, KAPPA achieves approximately 80-90% of LoRA's performance despite utilizing only a single-layer intervention with 32×16×4 times fewer parameters (Llama-2).
>
> This result serves as strong evidence for our hypothesis: a substantial portion of the model's errors stems from geometric misalignment between knowledge and prediction (which KAPPA addresses efficiently) rather than insufficient knowledge capacity (which requires LoRA's extensive parameters). If the problem were purely about capacity, KAPPA would have failed to achieve comparable performance with such minimal intervention.

---

> ### Author Response · Authors · 2025-12-03
> **Official Response to Reviewer (Part 2 of 5)**
>
> **A1 (continued).**
>
> **(4) Clarification on Performance Context**
>
> Regarding absolute task accuracy (though not our primary objective):
> - **vs. Inference-time baselines**: KAPPA substantially outperforms comparable inference-time methods—standard activation steering (+20.7% on BBH-Binary with Llama-2) and In-Context Learning (70.8% vs. 51.4%).
>
> - **vs. LoRA**: While KAPPA underperforms LoRA in absolute terms, this comparison requires context due to their fundamental differences. LoRA attaches adapters to every linear weight across all layers to inject new task-specific knowledge. KAPPA leaves the LLM completely frozen and intervenes only at a single layer to align predictions with existing knowledge. Since these methods address fundamentally different failure modes, applying KAPPA on top of a fine-tuned model could further reduce the knowledge-prediction gap—a promising direction for future work.
>
>
> **In summary,** our contribution is to (i) demonstrate the systematic knowledge-prediction gap, (ii) provide its geometric characterization, and (iii) show that simple calibration within the identified subspace can meaningfully adjust model behavior. Our results confirm that KAPPA operates exactly as designed to achieve these goals.
>
> ---
>
>
> >**W2.** Some unsupported claims are made during the analysis of the main-table results. l. 308: "these probes serve approximate upper bounds for each model’s achievable performance without any external information". This phrase makes several key claims which seem problematic. (a) "approximate upper bounds": the probe is clearly not the upper bound, since fine-tuning outperforms (b) "without external information": presumably this condition is provided to exclude fine-tuning from the claim, but I'm not sure this makes sense. "External information" is not defined, and its not clear how fine-tuning introduces more information than training probes does: they both leverage information from their training set.
>
> **A2.** We thank the reviewer for pointing out the ambiguity in our terminology. We agree that the phrasing "without external information" was imprecise. We have revised Section 4.2 to clearly define our terms as follows:
>
> **(1) Clarification on "Approximate Upper Bounds"**
>
> We acknowledge that this point was unclear. When we referred to the knowledge probe as an "approximate upper bound", we specifically meant the upper bound we can expect by reducing the knowledge-prediction gap using KAPPA. Since our main goal is to close this gap, the upper bound is naturally the accuracy of the knowledge that can be extracted from the frozen model, not the absolute performance limit achievable through further training (e.g., LoRA). Further training essentially injects external knowledge about questions and their correct answers by updating model parameters, so it can of course, achieve higher accuracy; however, this is not the objective of our work. The probe is trained on frozen representations, measuring the linearly extractable knowledge within the model's existing parameters—hence "approximate", as non-linear knowledge may not be fully captured. We therefore treat the probe accuracy as an approximate upper bound on the model's extractable knowledge without parameter updates.
>
> **(2) Correction on "External Information"**
>
> We acknowledge that the phrase "without any external information" was misleading, as training probes indeed requires external labels. We have therefore removed the statement and clarified the description in the revised manuscript.

---

> ### Author Response · Authors · 2025-12-03
> **Official Response to Reviewer (Part 3 of 5)**
>
> > **W3.** While results look better for free-form (since the probe is not applicable anymore), the results presented here are unclear. Which dataset is being tested on? Wouldn't it make more sense to test long-form QA tasks (e.g. summarization) in order to evaluate free-form response?
>
> **A3.** We apologize for the lack of clarity regarding the free-form experimental setup. We have revised the manuscript to explicitly include the following details.
>
> **(1) Dataset Specification: BBH Free-Form**
>
> As detailed in **Appendix J**, we constructed a free-form dataset based on **Big-Bench Hard (BBH).**
>
> - **Construction:** We systematically converted MCQ samples into open-ended questions. To ensure a valid evaluation, we applied keyword filtering to exclude questions that inherently require option selection (e.g., "Which statement best explains...").
> - **Scale:** This process yielded 1,759 valid free-form reasoning questions, serving as a rigorous testbed for the model's ability to generate correct reasoning chains without the aid of multiple-choice options.
>
> **(2) Why we chose this dataset**
>
> To clarify, free-form generation still relies on a knowledge-prediction subspace because we still calibrate the knowledge and prediction axes during the decoding process at the token level (detailed below). Hence, we chose a dataset for which we can obtain the knowledge-prediction subspace straightforwardly. For some open-ended questions, like summarization, although we could obtain a knowledge-prediction subspace using KAPPA as-is by binarizing the problems, determining the best way to binarize them requires further exploration. This is definitely an exciting research direction that can build on our work.
>
> In this work, we focused on the BBH dataset, for which we already have knowledge-prediction subspaces through binary-choice settings. Despite the simple and straightforward way to obtain these subspaces, the results (**accuracy improvement from 25.0% to 29.1%**) show that KAPPA generalizes beyond MCQ settings and remains effective in generative reasoning tasks. This demonstrates that calibrating the model within the identified subspace can enhance faithfulness to its internal knowledge even in free-form generation.
>
> **(3) Mechanism of KAPPA in Free-Form Generation**
>
> In this setting, KAPPA is not applied once to predict an answer symbol, but **continuously at each token generation step.**
>
> - For every token, the model's hidden state is projected onto the knowledge-prediction subspace.
> - KAPPA detects the mismatch between the knowledge basis and the prediction basis and dynamically adjusts the representation, thereby influencing the entire trajectory of generation.

---

> ### Author Response · Authors · 2025-12-03
> **Official Response to Reviewer (Part 4 of 5)**
>
> > **W4.** Justification for binarization of the datasets is never provided
>
> **A4.** We thank the reviewer for raising this issue, which meaningfully deepens our work. We agree that extending our method beyond the binary setting to handle 4+ choice questions is an important next step. Below, we elaborate on **how we can easily extend our method to multiple-choice settings** and demonstrate the result that our extended method consistently **yields the expected improvements on multiple-choice tasks**, following the trends observed in the original binary-choice setting.
>
> \
> **Extension to Multiple-Choice Settings.** We extend our method from binary to k-option MCQs by identifying geometric directions that encode how likely **each option** is correct (knowledge) and how likely the model is to choose each option (prediction), and then aligning hidden states along the directions.
>
> \
> **Probing.** For a $k$-option MCQ, we only make one change. We collect residual stream activations and train the knowledge and prediction probes as two single-layer **$k$-class** classifiers rather than binary classifiers. Specifically, given a hidden state  $h \in \mathbb{R}^{d \times 1}$,  $U_{\text{knowledge}}^{\top} h + b_{\text{knowledge}}$ and $U_{\text{prediction}}^{\top} h  + b_{\text{prediction}} \in \mathbb{R}^{k}$, where $U_{\text{knowledge}}, U_{\text{prediction}} \in \mathbb{R}^{d \times k}$ are weight matrices and $b_{\text{knowledge}}, b_{\text{prediction}} \in \mathbb{R}^{k \times 1}$ are biases, represent the logits for how likely each option is to be correct (knowledge) and how likely the model is to choose each option (prediction), respectively.
> These logits can be interpreted as coordinates in two $k$-dimensional subspaces of the residual stream: the knowledge subspace $\mathrm{span}(U_{\text{knowledge}})$ and the prediction subspace $\mathrm{span}(U_{\text{prediction}})$.
>
> \
> **Intervention.** Following the same principle as scalar coordinate alignment in the binary case, the multiple-choice setting extends this procedure by aligning the $k$-dimensional logits (rather than one-dimensional logits in the binary setting) produced by the knowledge and prediction probes. This can be interpreted as aligning each hidden state's coordinates in the prediction subspace with those in the knowledge subspace. To calibrate hidden states jointly along all k options, we solve the generalized version of the constrained optimization problem as in the binary case. This yields the generalized affine update:
> $$\tilde{h}' = \tilde{h} + \tilde{U}\_{\text{prediction}} (\tilde{U}\_{\text{prediction}}^{\top}\tilde{U}\_{\text{prediction}})^{-1} (\tilde{U}\_{\text{knowledge}}^{\top}\tilde{h} - \tilde{U}\_{\text{prediction}}^{\top}\tilde{h}),$$
> which reduces to the binary case when $k=1$.
> The update modifies the representation only within the prediction subspace, adjusting it to match the geometry of the representation in the knowledge subspace.
>
> \
> **Experiments.** We extend our evaluation to standard MCQ benchmarks with more than two options. Specifically, we use the 4-choice formulation of BBH (BBH-4C), the original 3-option ARC-Challenge (ARC-Challenge-3C), and the original 4-option MMLU (MMLU-4C).
>
> \
> **Results.** As shown in Table1 below, under this multiple-choice setting with increased difficulty and more complex geometry, the Llama-2-7B base model shows a substantial gap between the prediction accuracy and the knowledge probe’s accuracy. However, KAPPA **consistently reduces the gap**, indicating that KAPPA generalizes well beyond binary-choice to multiple-choice settings. Notably, on Llama-2, KAPPA yields substantial gains, improving **BBH-4C** accuracy by nearly **+10%** and **MMLU** accuracy by nearly **+5%**. For Qwen-2.5-7B, the improvements are smaller, which is expected because the original knowledge-prediction gap of the base model was smaller. Nevertheless, KAPPA effectively reduces the gap further. These results suggest that different models exhibit varying degrees of knowledge-prediction gap, while KAPPA consistently reduces it even in challenging multiple-choice scenarios.

---

> ### Author Response · Authors · 2025-12-03
> **Official Response to Reviewer (Part 5 of 5)**
>
> >**Q1.** l.393 "KAPPA’s success is not merely an artifact of manipulating binary choices but stems from an enhancement of the model’s ability to express its knowledge, indicating its potential to improve performance across a broader range of generative tasks". What do you mean by "manipulating binary choices"? This claim doesn't really connect with your previous argumentation that the fact that models will correctly answer in free-form and then mis-classify in a MCQA format, indicating prediction failure.
>
>
> **A1.** We thank the reviewer for pointing out this ambiguity. We appreciate the opportunity to clarify the intent behind this statement and how it connects to our broader argumentation **regarding the knowledge-prediction gap.**
>
> **(1) Clarification of "Manipulating Binary Choices"**
>
> By the phrase "merely an artifact of manipulating binary choices", we intended to clarify that KAPPA is not a mechanism tailored to a binary-choice setup, but rather a general intervention that adjusts the model’s internal representations to better reflect its latent knowledge. The method operates at the level of hidden-state geometry, not at the level of manipulating specific answer formats, and therefore applies beyond binary decisions.
>
> **(2) Connection to Previous Argumentation (The "General" Gap)**
>
> The statement "models correctly answer in free-form and then mis-classify in a MCQA format" is an example to demonstrate that models' internal knowledge and its expression can misalign depending on the answer format, particularly in a MCQ format in our work. Our claim that "KAPPA's success is not merely an …" aims to highlight that KAPPA can reduce this gap even in the free-form format. We have clarified this connection as below and incorporated it in Section 4.4 of the revision.

---

### Official Review · Reviewer_HJVi · 2025-10-22

**Soundness:** 2
**Presentation:** 2
**Contribution:** 3
**Rating:** 2
**Confidence:** 3

**Summary:**

This paper raises the issue of language models failing to answer question even when the correct information is contained within them. Authors apply probe analysis to the residual streams of transformer LLMs and discover the gap between subspaces representing the models actual knowledge and the answer it outputs for binary-choice questions; they show that linear probes from certain layers can achieve performance significantly higher than that of the model itself. Then, they propose an intervention technique that aligns the residual stream representations with the `knowledge' subspace while maximizing the preserved information.

Finally, authors apply their intervention techniques to two language models against competitive methods (activation steering, LoRA finetuning, etc.) and test them on several tasks, including binary question answering, free-form question answering, and general text generation. Results show that the proposed method noticeably improves the model's performance.

**Strengths:**

- A promising insight into the geometry of the model's "knowledge" in its internal representations.

- An interesting intervention method with well-developed mathematical background.

**Weaknesses:**

- The choice of title is somewhat misleading. Multiple-Choice Questions are explicitly stated in the title and abstract; however, the proposed method uses exclusively Binary choice questions to learn the transformation. Also, no evaluation on MCQA (questions with more than 2 options; for example, common MMLU) was reported in this paper. Can learning of the Knowledge-Aligned Prediction transformation be extended to general MCQA setup (e.g., 4-option questions)?

- Limited evaluation. Only two LLMs are explored, both of the same size (LLaMA2-7B and Qwen2.5-7B). It would be interesting to see how observed phenomena depend on the scale of the model; are there any qualitive changes. Also, some more recent models would've been much appreciated (LLaMA2 isn't a bad model, per se, but modern models usually perform much better on the reported benchmarks).

- Certain important details of the experimental setup are omitted.

	*) What is the size of train/validation/test subsets? Linear probes have quite a number of features (4096); what regularization was used to prevent overfitting (if any)?

 	*) What is the class balance in the Binary version of the datasets that were created/used in the experiments?

	*) For experiments on vicuna-eval (Section 3.4) could you provide more details (i.e., number of tasks per category, number of restarts for each prompt)? As of now, it is a little confusing. Reported overall performance doesn't seem to be an average of the performances for each category (46.9% vs. 45.2% for LoRA), and, at the same time, neither percentage of the number of tasks in the benchmark (80) is an integer.

- Reproducibility could not be assessed, because code wasn't provided.

**Questions:**

1. See Weaknesses

2. Please, make the fontsize on all figures larger.
3.  Please clarify in the caption for Table 2 whether you use -Binary versions of the datasets or not.
4. Could you please specify what model was used in the LLM-as-a-Judge setup for experiments with free-form generation?
5. Do you have any insights on how the proposed intervention affects the performance in the case of CoT or few-shot promptings?

---

> ### Author Response · Authors · 2025-12-03
> **Official Response to Reviewer (Part 1 of 4)**
>
> We thank the reviewer for their time in reviewing our paper and for recognizing our work as a promising insight. We also appreciate the positive evaluation of our method as interesting with a well-developed mathematical background.
>
> ---
>
> >  **W1.** The choice of title is somewhat misleading. Multiple-Choice Questions are explicitly stated in the title and abstract; however, the proposed method uses exclusively Binary choice questions to learn the transformation. Also, no evaluation on MCQA (questions with more than 2 options; for example, common MMLU) was reported in this paper. Can learning of the Knowledge-Aligned Prediction transformation be extended to general MCQA setup (e.g., 4-option questions)?
>
> **A1.** We thank the reviewer for the feedback regarding the scope of our title and experimental settings. We initially focused on the binary-choice setting as a simple starting point, and we fully agree that demonstrating this extension on standard multiple-choice benchmarks is important for completeness. Below, we elaborate on how we can **easily extend our method to multiple-choice settings** and demonstrate the result that our extended method consistently **yields the expected improvements** on multiple-choice tasks, following the trends observed in the original binary-choice setting.
>
> **Extension to Multiple-Choice Settings.** We extend our method from binary to k-option MCQs by identifying geometric directions that encode how likely **each option** is correct (knowledge) and how likely the model is to choose each option (prediction), and then aligning hidden states along the directions.
>
> **Probing.** For a $k$-option MCQ, we only make one change. We collect residual stream activations and train the knowledge and prediction probes as two single-layer **$k$-class** classifiers rather than binary classifiers. Specifically, given a hidden state  $h \in \mathbb{R}^{d \times 1}$,  $U_{\text{knowledge}}^{\top} h + b_{\text{knowledge}}$ and $U_{\text{prediction}}^{\top} h  + b_{\text{prediction}} \in \mathbb{R}^{k}$, where $U_{\text{knowledge}}, U_{\text{prediction}} \in \mathbb{R}^{d \times k}$ are weight matrices and $b_{\text{knowledge}}, b_{\text{prediction}} \in \mathbb{R}^{k \times 1}$ are biases, represent the logits for how likely each option is to be correct (knowledge) and how likely the model is to choose each option (prediction), respectively.
> These logits can be interpreted as coordinates in two $k$-dimensional subspaces of the residual stream: the knowledge subspace $\mathrm{span}(U_{\text{knowledge}})$ and the prediction subspace $\mathrm{span}(U_{\text{prediction}})$.
>
> **Intervention.** Following the same principle as scalar coordinate alignment in the binary case, the multiple-choice setting extends this procedure by aligning the $k$-dimensional logits (rather than one-dimensional logits in the binary setting) produced by the knowledge and prediction probes. This can be interpreted as aligning each hidden state's coordinates in the prediction subspace with those in the knowledge subspace. To calibrate hidden states jointly along all k options, we solve the generalized version of the constrained optimization problem as in the binary case. This yields the generalized affine update:
> $$\tilde{h}' = \tilde{h} + \tilde{U}\_{\text{prediction}} (\tilde{U}\_{\text{prediction}}^{\top}\tilde{U}\_{\text{prediction}})^{-1} (\tilde{U}\_{\text{knowledge}}^{\top}\tilde{h} - \tilde{U}\_{\text{prediction}}^{\top}\tilde{h}),$$
> which reduces to the binary case when $k=1$.
> The update modifies the representation only within the prediction subspace, adjusting it to match the geometry of the representation in the knowledge subspace.
>
> **Experiments.** We extend our evaluation to standard MCQ benchmarks with more than two options. Specifically, we use the 4-choice formulation of BBH (BBH-4C), the original 3-option ARC-Challenge (ARC-Challenge-3C), and the original 4-option MMLU (MMLU-4C).
>
> As shown in Table1 below, under this multiple-choice setting with increased difficulty and more complex geometry, the Llama-2-7B base model shows a substantial gap between the prediction accuracy and the knowledge probe’s accuracy. However, KAPPA **consistently reduces the gap**, indicating that KAPPA generalizes well beyond binary-choice to multiple-choice settings. Notably, on Llama-2, KAPPA yields substantial gains, improving **BBH-4C** accuracy by nearly **+10%** and **MMLU-4C** accuracy by nearly **+5%**. For Qwen-2.5-7B, the improvements are smaller, which is expected because the original knowledge-prediction gap of the base model is smaller. Nevertheless, KAPPA effectively reduces the gap further. These results suggest that different models exhibit varying degrees of knowledge-prediction gap, while KAPPA consistently reduces it even in challenging multiple-choice scenarios.

---

> ### Author Response · Authors · 2025-12-03
> **Official Response to Reviewer (Part 2 of 4)**
>
> **W1-A1 (continued).**
>
> *Table1: Results of Multiple-Choice Intervention on BBH-4C, MMLU-4C, and ARC-3C using Llama-2 7B Chat and Qwen2.5 7B Instruct. Numbers represent test accuracy percentages. KAPPA variants correspond to interventions applied to 1, 3 layers, respectively.*
> | Method          | Llama-2 BBH | Llama-2 MMLU | Llama-2 ARC | Qwen2.5 BBH | Qwen2.5 MMLU | Qwen2.5 ARC |
> |-----------------|--------------|--------------|--------------|--------------|----------------|--------------|
> | Base            | 27.6         | 38.8         | 62.5         | 53.0         | 69.3           | 89.5         |
> | KAPPA (1-layer) | 34.1         | 43.9           | 64.4         | 56.1         | 69.9           | 89.7         |
> | KAPPA (3-layer) | 36.9         | 44.5         | 63.8         | 55.4         | 69.9           | 89.6         |
> | Knowledge Probe | 44.4         | 45.2         | 64.5         | 57.7         | 70.3           | 89.7         |
>
> ---
>
> >**W2.** Limited evaluation. Only two LLMs are explored, both of the same size (LLaMA2-7B and Qwen2.5-7B). It would be interesting to see how observed phenomena depend on the scale of the model; are there any qualitive changes. Also, some more recent models would've been much appreciated (LLaMA2 isn't a bad model, per se, but modern models usually perform much better on the reported benchmarks).
>
> **A2.** We sincerely appreciate this review that adds depth to our research. To address whether our findings hold across different model scales and newer architectures, we have conducted additional experiments on **Qwen3-32B, Llama-3.1-8B, and Qwen3-4B** using BBH-Binary. Across diverse model families, **we observe substantial knowledge–prediction gaps**, and **KAPPA reliably reduces these gaps in every model**, bringing predictions significantly closer to the knowledge probe. Details are provided below.
>
> *Table 1. Results across multiple models on the BBH Binary benchmark, using the Instruct/Chat variants for all models. Columns for Llama-2 7B and Qwen2.5 7B are taken directly from the original paper.*
> | Method          | Qwen3 32B | Llama-3.1 8B | Llama-2 7B | Qwen2.5 7B | Qwen3 4B |
> | --------------- | --------- | ------------ | ---------- | ---------- | -------- |
> | Base            | 75.3      | 68.0         | 50.9       | 71.1       | 64.2     |
> | 1 Layer         | 78.4      | 80.7         | 69.8       | 72.3       | 76.2     |
> | 6 Layer         | 84.4      | 81.3         | 70.8       | 74.3       | 77.9     |
> | Knowledge Probe | 85.5      | 81.5         | 72.8       | 81.0       | 79.6     |
>
> **KAPPA consistently improves performance across all models**, with particularly larger gains on smaller models. As shown in the table below, KAPPA (6-layer) achieves +9.1% improvement on 32B, +13.3% on 8B, and +13.7% on 4B parameters. Notably, on Qwen3 4B, KAPPA boosts accuracy from 64.2% to 77.9%, nearly matching the knowledge probe (79.6%).
>
> **Knowledge–prediction gaps emerge across large, small, and newly released models**: substantial discrepancies between the model’s prediction and the knowledge probe persist regardless of scale. While the smaller 4B model’s base accuracy falls 15.4% short of the knowledge probe (64.2% vs. 79.6%), even the far larger 32B model exhibits a notable 10.2% gap (75.3% vs. 85.5%). More recent architectures, such as Llama-3.1 8B, show a substantial 13.5% descrepancy (68% vs. 81.5%).
>
> These results further highlight patterns of the knowledge-prediction gap across models: (i) Smaller models generally exhibit larger gaps (Qwen3 32B vs. Qwen3 4B), (ii) More recent architecture exhibits a smaller gap (Llama-2 7B vs. Llama-3.1 8B).  **KAPPA consistently closes these gaps across all models**, demonstrating strong scalability and broad applicability.
>
> ---
>
> >**W3.** Certain important details of the experimental setup are omitted.
>
> **A3.** We appreciate the reviewer's attention to these experimental details. We provide the requested information below:
>
> **Q. What is the size of train/validation/test subsets? Linear probes have quite a number of features (4096); what regularization was used to prevent overfitting (if any)?**
>
> For our experiments, we used the following train/validation/test splits:
>
> - BBH: 3,916 / 1,958 / 3,918 samples
> - MMLU: 9,498 / 4,786 / 9,582 samples
> - ARC: 2,238 / 598 / 2,344 samples
>
> Regarding regularization for the linear probes, we deliberately chose not to apply explicit regularization (e.g., L1/L2 penalties). This decision was motivated by two factors: (1) our relatively large training sets provide sufficient data points relative to the 4,096 features, yielding stable probe accuracies without overfitting signs, and (2) we wanted to obtain the most direct measurement of the knowledge encoded in the representations without introducing additional hyperparameters that could affect comparability across different model layers and tasks.

---

> ### Author Response · Authors · 2025-12-03
> **Official Response to Reviewer (Part 3 of 4)**
>
> **W3-A3 (continued).**
>
> **Q.What is the class balance in the Binary version of the datasets that were created/used in the experiments?**
>
> Our binary versions maintain perfect class balance (50%:50%) by design. Each question is evaluated with both option orders (e.g., "(A) correct, (B) incorrect" and "(A) incorrect, (B) correct"), ensuring that any position bias is neutralized while maintaining exact class balance across the dataset.
>
> **Q.For experiments on vicuna-eval (Section 3.4) could you provide more details (i.e., number of tasks per category, number of restarts for each prompt)? As of now, it is a little confusing. Reported overall performance doesn't seem to be an average of the performances for each category (46.9% vs. 45.2% for LoRA), and, at the same time, neither percentage of the number of tasks in the benchmark (80) is an integer.**
>
> We apologize for the confusion regarding the Vicuna-eval numbers.
>
> Here's a clarification:
>
> The benchmark contains 80 unique questions across 9 categories. However, to ensure robust evaluation and mitigate position bias, we employ a two-fold evaluation protocol:
> 1. Each question is evaluated twice with swapped response orders (Model A vs. B, then B vs. A)
> 2. This results in 160 total comparisons (80 questions × 2 orders)
> 3. Each evaluation starts fresh as a first-turn message in a new session
>
> *Table 1. Distribution of question categories.*
> | Category        | Count |
> |-----------------|-------|
> | generic         | 20    |
> | knowledge       | 20    |
> | roleplay        | 20    |
> | common-sense    | 20    |
> | fermi           | 20    |
> | counterfactual  | 20    |
> | coding          | 14    |
> | math            | 6     |
> | writing         | 20    |
> | TOTAL       | 160 |
>
> The overall win rate (46.9% for KAPPA vs. 45.2% for LoRA) is computed across all 160 comparisons. The slight discrepancy from category-wise averages occurs because categories have different numbers of questions (e.g., coding has 7, math has 3), making the overall rate a weighted average rather than a simple mean. This evaluation protocol ensures that our comparisons are robust to both position bias and category imbalance, providing a fair assessment of each method's performance in generating preferred responses.
>
> These clarifications have been added to Appendix E.1, which details the data splits used in the binary-choice evaluation, and Appendix N.1, which details the question categories used for the free-form evaluation in the revised manuscript.
>
> ---
>
> >**W4.** Reproducibility could not be assessed, because code wasn't provided.
>
> **A4.** We will release our code publicly upon publication.
>
> ---
>
> >**Q2, Q3.** 2. Please, make the fontsize on all figures larger. 3. Please clarify in the caption for Table 2 whether you use -Binary versions of the datasets or not.
>
> **A2, A3.** Thank you for these suggestions. We have:
> Increased font sizes in all figures (especially Fig. 3, 7, 8)
> Clarified Table 2 caption to explicitly state "-Binary versions are used for all datasets"
>
> ---
>
> >**Q4.** Could you please specify what model was used in the LLM-as-a-Judge setup for experiments with free-form generation?
>
> **A4.** We thank the reviewer for pointing out this missing information. To ensure rigorous evaluation, we utilized the following models for our LLM-as-a-Judge setups:
> - **Free-form Generation (Section 4.3):** We used **GPT-4o-mini** (version: gpt-4o-mini-2025-04-16) to evaluate whether the ground-truth answer is semantically contained within or equivalent to the model's generated response.
> - **General Capability (Section 4.4):** As briefly mentioned in line 409, we used **GPT-5** (reasoning effort: high) for the vicuna-eval benchmark to ensure high-quality assessment of general instruction-following capabilities.
>
> We have explicitly included these model specifications and version details in the Appendix of the final manuscript.
>
> ---
>
> >**Q5.** Do you have any insights on how the proposed intervention affects the performance in the case of CoT or few-shot promptings?
>
> **A5.** We thank the reviewer for this insightful question. While our current experiments focused on zero-shot settings to isolate the baseline knowledge-prediction gap, we expect KAPPA holds promise for Chain-of-Thought (CoT) and few-shot scenarios based on our findings in generative tasks.

---

> ### Author Response · Authors · 2025-12-03
> **Official Response to Reviewer (Part 4 of 4)**
>
> **Q5-A5 (continued).**
>
> **(1) Evidence from Free-Form Tasks**
>
> Our experiments confirm that KAPPA is not limited to single-token prediction but also improves sequential generation. In the free-form variant of BBH, KAPPA increases accuracy from 25.0% to 29.1%, demonstrating that the intervention enhances faithfulness to the model’s internal knowledge even in multi-token generation settings.
>
> Although free-form generation operates over a large vocabulary, individual decoding steps still involve selecting among **a small set of plausible next-token candidates**, where the knowledge–prediction gap may arise. The observed gains suggest that KAPPA effectively corrects these misaligned token-level decisions.
>
> **(2) Hypothesis for CoT: Mitigating Error Propagation**
>
> Since KAPPA intervenes at every token during generation, it naturally affects the intermediate reasoning process. We hypothesize that KAPPA helps **make key token-level decisions more faithful at each step**, thereby mitigating the propagation of early reasoning errors throughout the chain of thought.
>
> **(3) Hypothesis for Few-Shot Prompting: Complementary Roles**
>
> We hypothesize that the two approaches play complementary roles. Few-shot prompting **enriches the model’s internal evidence** by providing task-relevant demonstrations in the input context, thereby strengthening the knowledge encoded during inference. In contrast, KAPPA ensures that the model’s predictions remain **faithfully grounded in that knowledge.**

---

### Official Review · Reviewer_jtgY · 2025-10-27

**Soundness:** 3
**Presentation:** 3
**Contribution:** 2
**Rating:** 6
**Confidence:** 4

**Summary:**

This paper introduces a method to address the knowledge-prediction gap in Large Language Models (LLMs) on multiple-choice questions, called KAPPA. The authors identify a two-dimensional subspace within LLM hidden states spanned by: (1) a knowledge basis encoding the probability of the correct answer, and (2) a prediction basis encoding the probability of the model's predicted answer. KAPPA intervenes at inference time by geometrically realigning these coordinates to make predictions more faithful to internal knowledge. The method improves LLMs performance across several datasets and extends to free-form generation.

**Strengths:**

- The paper provides a clear intuitive framework for understanding the knowledge-prediction gap through a two-dimensional subspace representation;
- The authors conduct extensive experiments including cross-dataset transfer, free-form generation, and general capability assessment;
- The paper is well-written and the main idea is easy to understand.

**Weaknesses:**

- The current method only handles binary-choice questions and don't consider more general MCQ scenarios with 4+ options.
- While the geometric framework is intuitive, the paper doesn't deeply explain why this particular misalignment occurs or what causes it during training. Understanding the root cause could lead to better solutions.
- The paper does not cite "Listening to the Wise Few: Select-and-Copy Attention Heads for Multiple-Choice QA" (Tulchinskii et al., 2024), which analyzed - prior to this work - the phenomenon that LLMs can encode correct knowledge internally yet fail to express it in MCQ predictions. This oversight misrepresents the novelty of the core observation and suggests incomplete literature review.
- Experiments focus on only two 7B parameter models (LLaMA-2 and Qwen-2.5). I've worked with LLaMA 2 and observed clear differences in its performance on MCQ tasks compared with newer models. A newer models need to be included into this research. Also it would be good to check the models of more various sizes (less than 7B, more than 7B).

**Questions:**

- How does the method scale to multiple-choice questions with >2 options? Is there any barrier to try these scenarios?
- Have you investigated what properties of training data or procedures lead to this knowledge-prediction misalignment?
- Could this approach be combined with training-time interventions to prevent the gap from forming initially?
- How does performance vary with model scale? Do larger models exhibit smaller knowledge-prediction gaps?

---

> ### Author Response · Authors · 2025-12-03
> **Official Response to Reviewer (Part 1 of 4)**
>
> Thank you for taking the time to read our paper and for providing such constructive feedback. We are genuinely grateful that you found our framework clear and intuitive. We also appreciate your recognition of the scope of our experimental evaluation and your positive remarks on the writing.
>
> ---
>
> >  **W1.** The current method only handles binary-choice questions and don't consider more general MCQ scenarios with 4+ options.
>
> **A1.** We thank the reviewer for raising this issue, which meaningfully deepens our work. We agree that extending our method beyond the binary setting to handle 4+ choice questions is an important next step. Below, we elaborate on **how we can easily extend our method to multiple-choice settings** and demonstrate the result that our extended method consistently **yields the expected improvements on multiple-choice tasks**, following the trends observed in the original binary-choice setting.
>
> **Extension to Multiple-Choice Settings.** We extend our method from binary to k-option MCQs by identifying geometric directions that encode how likely **each option** is correct (knowledge) and how likely the model is to choose each option (prediction), and then aligning hidden states along the directions.
>
> **Probing.** For a $k$-option MCQ, we only make one change. We collect residual stream activations and train the knowledge and prediction probes as two single-layer **$k$-class** classifiers rather than binary classifiers. Specifically, given a hidden state  $h \in \mathbb{R}^{d \times 1}$,  $U_{\text{knowledge}}^{\top} h + b_{\text{knowledge}}$ and $U_{\text{prediction}}^{\top} h  + b_{\text{prediction}} \in \mathbb{R}^{k}$, where $U_{\text{knowledge}}, U_{\text{prediction}} \in \mathbb{R}^{d \times k}$ are weight matrices and $b_{\text{knowledge}}, b_{\text{prediction}} \in \mathbb{R}^{k \times 1}$ are biases, represent the logits for how likely each option is to be correct (knowledge) and how likely the model is to choose each option (prediction), respectively.
> These logits can be interpreted as coordinates in two $k$-dimensional subspaces of the residual stream: the knowledge subspace $\mathrm{span}(U_{\text{knowledge}})$ and the prediction subspace $\mathrm{span}(U_{\text{prediction}})$.
>
> **Intervention.** Following the same principle as scalar coordinate alignment in the binary case, the multiple-choice setting extends this procedure by aligning the $k$-dimensional logits (rather than one-dimensional logits in the binary setting) produced by the knowledge and prediction probes. This can be interpreted as aligning each hidden state's coordinates in the prediction subspace with those in the knowledge subspace. To calibrate hidden states jointly along all k options, we solve the generalized version of the constrained optimization problem as in the binary case. This yields the generalized affine update:
> $$\tilde{h}' = \tilde{h} + \tilde{U}\_{\text{prediction}} (\tilde{U}\_{\text{prediction}}^{\top}\tilde{U}\_{\text{prediction}})^{-1} (\tilde{U}\_{\text{knowledge}}^{\top}\tilde{h} - \tilde{U}\_{\text{prediction}}^{\top}\tilde{h}),$$
> which reduces to the binary case when $k=1$.
> The update modifies the representation only within the prediction subspace, adjusting it to match the geometry of the representation in the knowledge subspace.
>
> **Experiments.** We extend our evaluation to standard MCQ benchmarks with more than two options. Specifically, we use the 4-choice formulation of BBH (BBH-4C), the original 3-option ARC-Challenge (ARC-Challenge-3C), and the original 4-option MMLU (MMLU-4C).
>
> **Results.** As shown in Table1 below, under this multiple-choice setting with increased difficulty and more complex geometry, the Llama-2-7B base model shows a substantial gap between the prediction accuracy and the knowledge probe’s accuracy. However, KAPPA **consistently reduces the gap**, indicating that KAPPA generalizes well beyond binary-choice to multiple-choice settings. Notably, on Llama-2, KAPPA yields substantial gains, improving **BBH-4C** accuracy by nearly **+10%** and **MMLU** accuracy by nearly **+5%**. For Qwen-2.5-7B, the improvements are smaller, which is expected because the original knowledge-prediction gap of the base model was smaller. Nevertheless, KAPPA effectively reduces the gap further. These results suggest that different models exhibit varying degrees of knowledge-prediction gap, while KAPPA consistently reduces it even in challenging multiple-choice scenarios.

---

> ### Author Response · Authors · 2025-12-03
> **Official Response to Reviewer (Part 2 of 4)**
>
> **W1-A1 (continued).**
>
> *Table1: Results of Multiple-Choice Intervention on BBH-4C, MMLU-4C, and ARC-3C using Llama-2 7B Chat and Qwen2.5 7B Instruct. Numbers represent test accuracy percentages. KAPPA variants correspond to interventions applied to 1, 3 layers, respectively.*
> | Method          | Llama-2 BBH | Llama-2 MMLU | Llama-2 ARC | Qwen2.5 BBH | Qwen2.5 MMLU | Qwen2.5 ARC |
> |-----------------|--------------|--------------|--------------|--------------|----------------|--------------|
> | Base            | 27.6         | 38.8         | 62.5         | 53.0         | 69.3           | 89.5         |
> | KAPPA (1-layer) | 34.1         | 43.9           | 64.4         | 56.1         | 69.9           | 89.7         |
> | KAPPA (3-layer) | 36.9         | 44.5         | 63.8         | 55.4         | 69.9           | 89.6         |
> | Knowledge Probe | 44.4         | 45.2         | 64.5         | 57.7         | 70.3           | 89.7         |
>
> ---
>
> >  **W2.** While the geometric framework is intuitive, the paper doesn't deeply explain why this particular misalignment occurs or what causes it during training. Understanding the root cause could lead to better solutions.
>
> **A2.** We appreciate the reviewer’s thoughtful comment. We outline below several plausible factors contributing to the knowledge-prediction gap, informed by prior work and our own observations.
>
> **Distracting Mechanisms:** First, many studies (including our own analyses) indicate that correct knowledge is predominantly encoded in earlier layers, while later layers largely determine the final prediction [1-3]. Prior MCQ research has also shown that attention heads encoding correct knowledge and those introducing biases tend to be sparsely separated [3-5]. This suggests that models often **possess mechanisms that identify the correct answer**, but could be overridden by competing mechanisms that influence output prediction over layers. The gap may emerge when these **distracting mechanisms dominate the final prediction**. Training on limited or biased supervised datasets can reinforce the use of superficial cues (e.g., option position, length, or stylistic markers)  [6-9]. Consistent with this view, our layerwise analyses show that the knowledge signal is already salient in middle layers, whereas the misalignment between the knowledge and prediction bases emerges primarily in later layers where symbol-level biases become prominent.
>
> **Layer-level shifts:** Prior work notes that **post-training significantly alters mid-to-late layers**, while leaving earlier knowledge-encoding layers relatively unchanged. This shift can cause **miscalibration between where knowledge is stored and where predictions are generated** [10]. This may cause the model to retain correct knowledge in early layers yet produce misaligned predictions in later layers. Consistent with this observation, our work demonstrates that the model indeed forms distinct one-dimensional subspaces for knowledge encoding and prediction encoding, but these directions are systematically miscalibrated. We further show that a simple geometric calibration between the two effectively resolves this mismatch, revealing a closely related underlying mechanism.
>
> A deeper investigation into the underlying causes of this gap is a valuable direction but would require a dedicated effort in future research. We have incorporated this discussion in the related work section of the revision.
>
> **References:**
> - [1] Boiko et al. (2025). Estimating Knowledge in Large Language Models Without Generating a Single Token. EMNLP 2024.
> - [2] Wiegreffe et al. (2025). Answer, Assemble, Ace: Understanding How LMs Answer Multiple Choice Questions. ICLR 2025.
> - [3] Yang et al. (2025). Option Symbol Matters: Investigating and Mitigating Multiple-Choice Option Symbol Bias of Large Language Models. NAACL 2025.
> - [4] Tulchinskii et al. (2024). Listening to the Wise Few: Select-and-Copy Attention Heads for Multiple-Choice QA. arXiv preprint, 2024.
> - [5] Li & Gao (2025). Anchored Answers: Unravelling Positional Bias in GPT-2’s Multiple-Choice Questions. Findings of ACL 2025.
> - [6] Pezeshkpour & Hruschka (2024). LLMs’ Sensitivity to the Order of Options in MCQs. Findings of NAACL 2024.
> - [7] Zheng et al. (2023). Large language models are not robust multiple choice selectors. arXiv preprint, 2023.
> - [8] Zhao et al. (2025). LLMs badly generalize across option length and problem types. EMNLP 2025.
> - [9] Xue et al. (2024). Strengthened symbol binding makes LLMs reliable MCQ selectors. ACL 2024.
> - [10] Gottesman & Geva (2024). Estimating Knowledge in LLMs Without Generating a Single Token. EMNLP 2024.

---

> ### Author Response · Authors · 2025-12-03
> **Official Response to Reviewer (Part 3 of 4)**
>
> >**W3.** The paper does not cite "Listening to the Wise Few: Select-and-Copy Attention Heads for Multiple-Choice QA" (Tulchinskii et al., 2024), which analyzed - prior to this work - the phenomenon that LLMs can encode correct knowledge internally yet fail to express it in MCQ predictions. This oversight misrepresents the novelty of the core observation and suggests incomplete literature review.
>
> **A3.** We thank the reviewer for this important reference. We have cited Tulchinskii et al. (2024) and discussed it in the related work section of the revised manuscript. Below, we clarify that while both the reference and our work examine the knowledge-prediction gap, they differ in their analytical focus, experimental scope, and underlying motivations.
>
> First, Tulchinskii et al. and our work share the observation that LLMs can internally encode correct knowledge yet fail to express it in MCQ predictions. Both works address this phenomenon through mechanistic interpretability approaches by analyzing internal model states.
>
> However, there are two key differences between these approaches.
>
> 1. **Scope and generalizability.** Tulchinskii et al. focus specifically on improving MCQ performance through format-specific mechanisms. In contrast, we view MCQ as a testbed to study a broader phenomenon: the knowledge-prediction gap that manifests across diverse task formats. **We empirically demonstrate this generality**: our discovered basis vectors transfer across different datasets (Table 2) and task formats, including free-form QA (improves 25.0% to 29.1%). This cross-format consistency suggests KAPPA captures fundamental properties of knowledge representation rather than task-specific patterns.
> 2. **Theoretical contribution.** While Tulchinskii et al. provide valuable insights into which components perform answer selection, **KAPPA offers a geometric framework** that explains how knowledge is encoded and why misalignment occurs. Our 2D subspace structure provides a quantitative, interpretable model of the gap. This framework enables us to measure divergence between the internal knowledge and the prediction and correct it systematically, rather than bypassing the model's final output.
> 3. **Component of analysis and intervention mechanism.** Tulchinskii et al. operate at the attention mechanism level, identifying specific "select-and-copy" heads through validation set calibration and using their QK-scores for prediction. **KAPPA operates at the residual stream**, defining the knowledge-prediction gap as misalignment between two 1D basis vectors within the residual stream. This enables (a) closed-form optimization without validation set calibration, (b) a principled theoretical upper bound (knowledge probe accuracy), and (c) systematic correction rather than bypassing later model layers.
>
> These works offer complementary perspectives on the same phenomenon, and together contribute to a richer understanding of how LLMs process and express knowledge. We have  added an appropriate discussion of this work in the related work of our revision.
>
> ---
>
> >**W4.** Experiments focus on only two 7B parameter models (LLaMA-2 and Qwen-2.5). I've worked with LLaMA 2 and observed clear differences in its performance on MCQ tasks compared with newer models. A newer models need to be included into this research. Also it would be good to check the models of more various sizes (less than 7B, more than 7B).
>
> **A4.** We sincerely appreciate this review that adds depth to our research. To address whether our findings hold across different model scales and newer architectures, we have conducted additional experiments on **Qwen3-32B, Llama-3.1-8B, and Qwen3-4B** using BBH-Binary. Across diverse model families, **we observe substantial knowledge–prediction gaps**, and **KAPPA reliably reduces these gaps in every model**, bringing predictions significantly closer to the knowledge probe. Details are provided below.
>
> *Table 1. Results across multiple models on the BBH Binary benchmark, using the Instruct/Chat variants for all models. Columns for Llama-2 7B and Qwen2.5 7B are taken directly from the original paper.*
> | Method          | Qwen3 32B | Llama-3.1 8B | Llama-2 7B | Qwen2.5 7B | Qwen3 4B |
> | --------------- | --------- | ------------ | ---------- | ---------- | -------- |
> | Base            | 75.3      | 68.0         | 50.9       | 71.1       | 64.2     |
> | 1 Layer         | 78.4      | 80.7         | 69.8       | 72.3       | 76.2     |
> | 6 Layer         | 84.4      | 81.3         | 70.8       | 74.3       | 77.9     |
> | Knowledge Probe | 85.5      | 81.5         | 72.8       | 81.0       | 79.6     |

---

> ### Author Response · Authors · 2025-12-03
> **Official Response to Reviewer (Part 4 of 4)**
>
> **W4-A4 (continued).**
>
>
> **KAPPA consistently improves performance across all models**, with particularly larger gains on smaller models. As shown in the table below, KAPPA (6-layer) achieves +9.1% improvement on 32B, +13.3% on 8B, and +13.7% on 4B parameters. Notably, on Qwen3 4B, KAPPA boosts accuracy from 64.2% to 77.9%, nearly matching the knowledge probe (79.6%).
>
> **Knowledge–prediction gaps emerge across large, small, and newly released models**: substantial discrepancies between the model’s prediction and the knowledge probe persist regardless of scale. While the smaller 4B model’s base accuracy falls 15.4% short of the knowledge probe (64.2% vs. 79.6%), even the far larger 32B model exhibits a notable 10.2% gap (75.3% vs. 85.5%). More recent architectures, such as Llama-3.1 8B, show a substantial 13.5% descrepancy (68% vs. 81.5%).
>
> These results further highlight patterns of the knowledge-prediction gap across models: (i) Smaller models generally exhibit larger gaps (Qwen3 32B vs. Qwen3 4B), (ii) More recent architecture exhibits a smaller gap (Llama-2 7B vs. Llama-3.1 8B).  **KAPPA consistently closes these gaps across all models**, demonstrating strong scalability and broad applicability.
>
> ---
>
> >**Q1.** How does the method scale to multiple-choice questions with >2 options? Is there any barrier to try these scenarios?
>
> **A1.** As noted in our response to W1, we extend the method and report results on our multiple-choice evaluations, where our method yields consistent improvements for 3- and 4-choice questions.
>
> ---
>
> >**Q2.** Have you investigated what properties of training data or procedures lead to this knowledge-prediction misalignment?
>
> **A2.** Please see our response to W2, where we discuss both the mechanism underlying the knowledge-prediction gap and the specific training data and procedural factors that may contribute to this misalignment.
>
> ---
>
> >**Q3.** Could this approach be combined with training-time interventions to prevent the gap from forming initially?
>
> **A3.** We thank the reviewer for raising this important question. We believe that incorporating our approach directly into training is a promising direction. One potential strategy is to introduce an alignment loss as a joint training objective. After identifying the knowledge and prediction bases in the base model, the loss encourages the prediction coordinate to remain consistent with the knowledge coordinates throughout training. Such a formulation may further reduce the knowledge–prediction gap without compromising the model’s overall performance. We view this as an exciting direction for future work.
>
> ---
>
> >**Q4.** How does performance vary with model scale? Do larger models exhibit smaller knowledge-prediction gaps?
>
> **A4.** Please refer to our response to W4, where we discuss this question in detail. In summary, we observe that smaller models tend to exhibit larger knowledge–prediction gaps, and KAPPA effectively reduces this gap across all model scales.

---

### Official Review · Reviewer_ePcc · 2025-10-31

**Soundness:** 2
**Presentation:** 2
**Contribution:** 2
**Rating:** 2
**Confidence:** 3

**Summary:**

The authors introduce a new probing technique that dynamically updates the residual stream of an LLM. Then show that it can be used to improve outcomes on multiple choice benchmarks

**Strengths:**

The authors show that their method improves LLM performance on some benchmarks, and that the intervention appears to generalize slightly. I think the steering intervention as a correction in a learned plane is neat and should be explored more. The authors do a good job of exploring uses for their steering methods and are precise with describing its impact and limitations.

## originality

The steering vector contributions are novel as is the KAPPA technique, but both are directly building on a large body of existing techinques

 ## quality

The analysis and methods look sound, but the tests are limited to a few models and I still have questions about specific parts of the experiments

 ## clarity

I found this readable, but I found determining what exactly happened and what the impact was to be difficult. For example to me table 1 is of major importance for evaluating the results, and understanding if KAPPA generalizes. Showing each entry as the difference from the baseline would make reading it much easier.

## significance

I'm not convinced this method will work in general, so cannot say this is very significant, despite being intriguing

**Weaknesses:**

I don't think the method is as powerful as the authors claim. This feels like a different type of fine tuning (maybe a simple LORA), so the small boost in performance makes sense as you're "fine-tuning" general purpose models to answer questions better. I don't think it shows the claimed Knowledge dimension. I'm also skeptical of the claims this will generalize, either improving other benchmarks or work for more complex tasks (binary - -> multiple- choice for example), as they will require significantly more complex subspace. As an example multi-calibration is a very powerful technique that allows for similar strong claims about improving results via post processing, but the techniques are hamper if the number of classes that are being calibrated across get above certain thresholds. If I'm wrong about this please let me know.

**Questions:**

Could you explain more clearly what the "prediction basis" is? The description "encodes the probability of the model’s own choice" sounds  like a tautology. More generally I find the anthropomorphic language used in the paper to describe the models makes understanding more difficult.

In the intro line 47 you make the claim "We find that in the residual streams of certain layers, there exists a subspace spanned by two functionally distinct bases", can you explain how this proves there exists this subspace. And do you mean it exists in the LLM? The probes? or as a platonic form?

Can this be interpreted as a very low rank LoRA?

---

> ### Author Response · Authors · 2025-12-03
> **Official Response to Reviewer (Part 1 of 5)**
>
> We sincerely appreciate the reviewer for taking the time to provide a thoughtful and careful assessment of our work. We are grateful that you found our steering formulation, which interprets the intervention as a correction applied within a learned subspace, to be a neat and meaningful perspective.
>
> >**W1.** I don't think the method is as powerful as the authors claim. This feels like a different type of fine tuning (maybe a simple LORA), so the small boost in performance makes sense as you're "fine-tuning" general purpose models to answer questions better. I don't think it shows the claimed Knowledge dimension.
>
> **A1.** We apologize for the lack of clarity. To address your points, in the sections below, we clarify:
>
> - **(1) why we claimed the "knowledge dimension" captures the model's internal knowledge:** The knowledge probe we derive from the model’s residual stream predicts correct answers far more accurately than the base LLM’s natural predictions. This indicates that the probe substantially encodes the model’s internal knowledge. Since this probe forms a basis vector of the knowledge-prediction subspace, we state that the knowledge basis captures the model knowledge.
>
> - **(2) how our work differs from fine-tuning:** We freeze all parameters and adjust the hidden representations only at inference time, aligning predictions with existing knowledge encoded in the model rather than introducing new information.
>
> - **(3) in what sense our method is "powerful":** Our method is not designed to boost task accuracy but to close the knowledge-prediction gap—a property that our experiments consistently validate.
>
> These clarifications have been incorporated into the uploaded revision in both the introduction and method sections.
>
> ---
>
> **(1) Why the claimed "knowledge basis" captures the model's knowledge**
>
> We first obtain the knowledge basis and the prediction basis by training two separate linear classifiers (logistic regressions) on the model’s hidden representations. Concretely, for each question, we extract the residual-stream activation at a specific layer, denoted by $h$, together with two types of labels: (i) the ground-truth answer label, and (ii) the LLM’s own predicted response (e.g.,1 for option A and 0 for option B). We then train two linear classifiers with distinct objectives:
> - **Knowledge probe:** a classifier trained to predict the ground-truth label from $h$.
> - **Prediction probe:** a classifier trained to predict the model’s output response from $h$.
>
> Each classifier has parameters ($u$, $b$) and is trained so that *sigmoid*($u^\top h + b$) matches its corresponding label (ground-truth or model prediction). After training, **the resulting weight vectors** $u_{\text{knowledge}}$ and $u_{\text{prediction}}$ define the knowledge basis and the prediction basis, respectively.
>
> For the knowledge basis $u_{\text{knowledge}}$, a larger value of $u_{\text{knowledge}}^\top h$ represents a higher probability of option “A” being the correct answer **indicated by the model's hidden representation.** The reason we call $u_{\text{knowledge}}$ "knowledge basis" is twofold.
>
> - (1) Empirically, the calculated probability has quite high accuracy, substantially higher than both the base LLM's answer choices and the probability calculated using the prediction basis. This suggests that a significant amount of latent knowledge is encoded in the hidden states and can be captured by $u_{\text{knowledge}}$. Of course, we acknowledge that $u_{\text{knowledge}}$ is likely to capture the model's knowledge to a certain degree, but not perfectly, and serves as a lower bound.
> - (2) $u_{\text{knowledge}}$ can be interpreted as an axis in a subspace of the residual space (here, the residual space is a d-dimensional space where d is the dimensionality of the residual hidden states), where $u_{\text{knowledge}}^\top h$ is a projection of the hidden state onto this axis (i.e., a coordinate value). This is why we use the term "knowledge basis".

---

> ### Author Response · Authors · 2025-12-03
> **Official Response to Reviewer (Part 2 of 5)**
>
> **W1-A1(continued).**
>
> **(2) How our method fundamentally differs from fine-tuning**
>
> Our approach differs from fine-tuning in both its **mechanism** and its core **objective**:
> - **Difference in Mechanism: Updating Model Parameters vs. Probing-Intervention Paradigm**
>
>   Our method keeps the LLM entirely frozen and trains only two linear probes. Rather than updating any LLM parameters, we train the probes to localize a 2D subspace in the activation space where the base LLM consistently encodes information relevant to solving the task. No gradient flows through the LLM.
>
>   Methodologically, our approach follows the inference-time intervention paradigm, where prior work similarly trains linear probes to identify meaningful directions in hidden space and then modifies hidden representations during inference [1-4].
>
> - **Difference in Purpose: Knowledge Injection vs. Faithfulness Alignment**
>
>   Fine-tuning aims to modify the LLM by injecting new task-specific knowledge. KAPPA has a fundamentally different purpose: we do not inject new information into the model. Instead, we use training data only to **identify a 2D subspace** in which the model encodes (i) the latent knowledge supporting the correct answer and (ii) the mechanism driving its actual prediction.
>
>   Our objective is to make the LLM’s predictions more **faithful** to the hidden knowledge **already** encoded in the hidden states. While this calibration typically improves the model's task accuracy, as shown in our experiments, our objective goes beyond mere accuracy improvements. Rather, the more important goals we aim to achieve using KAPPA (and our main contributions) are: (i) demonstrating the presence of a systematic knowledge–prediction gap, (ii) providing a geometric characterization of this gap, and (iii) showing that a simple calibration within the identified subspace can meaningfully adjust model behavior. Our goal is to motivate further research on understanding and bridging this structural misalignment.
>
> ---
>
> **(3) In what sense our method is powerful**
>
> As discussed above, it should be clarified that KAPPA was not primarily designed to improve task accuracy but rather to **close the knowledge-prediction gap.** Hence, its "powerfulness" should be assessed in that regard. We demonstrated this in at least two ways.
>
> First, the calibration using KAPPA makes the model’s answer choice to become more aligned with the knowledge probe. This alignment is quantitatively reflected in the agreement rate between the knowledge probe’s predicted answers and the model’s actual outputs (See Table1 below). Second, as shown in Figure~4a, LoRA fine-tuning still leaves a 5.7% rate of “correct in free-form, incorrect in MCQ” cases, whereas KAPPA reduces this gap to 5.1%. Given that free-form questions elicit the model's knowledge more naturally, as they are not constrained like multiple-choice questions, this result demonstrates KAPPA's effectiveness in reducing the knowledge-prediction gap.
>
> *Table 1: Agreement Rate (%) on Llama-2 7B Chat (Layer 17) and Qwen2.5 7B Instruct (Layer 20)).*
> | Dataset      | Llama-2 Base | Llama-2 KAPPA | Qwen-2.5 Base | Qwen-2.5 KAPPA |
> |--------------|--------------|----------------|----------------|-----------------|
> | BBH-Binary   | 0.508        | 0.839          | 0.710          | 0.781           |
> | MMLU-Binary  | 0.641        | 0.800          | 0.808          | 0.885           |
> | ARC-Binary   | 0.742        | 0.916          | 0.919          | 0.977           |
>
> **References:**
> - [1] Li et al. (2023). Inference-time intervention: eliciting truthful answers from a language model. NeurIPS 2023.
> - [2] Zhang et al. (2024). TruthX: Alleviating Hallucinations by Editing Large Language Models in Truthful Space. ACL 2024.
> - [3] Rütte et al. (2024). A language model's guide through latent space. ICML 2024.
> - [4] Chen et al. (2024). Truth Forest: Toward multi-scale truthfulness in large language models through intervention without tuning. AAAI 2024.
> ---
> >**W2.** I'm also skeptical of the claims this will generalize, either improving other benchmarks or work for more complex tasks (binary - -> multiple- choice for example), as they will require significantly more complex subspace. As an example multi-calibration is a very powerful technique that allows for similar strong claims about improving results via post processing, but the techniques are hamper if the number of classes that are being calibrated across get above certain thresholds.
>
> **A2.** Thank you for pointing this out. We agree that extension to multiple-choice settings with more choices is an important direction. Below, we elaborate on **how we can easily extend our method to multiple-choice settings** and demonstrate the result that our extended method consistently **yields the expected improvements on multiple-choice tasks**, following the trends observed in the original binary-choice setting.

---

> ### Author Response · Authors · 2025-12-03
> **Official Response to Reviewer (Part 3 of 5)**
>
> **W2-A2 (continued).**
>
> **Extension to Multiple-Choice Settings.** We extend our method from binary to k-option MCQs by identifying geometric directions that encode how likely **each option** is correct (knowledge) and how likely the model is to choose **each option** (prediction), and then aligning hidden states along the directions.
>
> **Probing.** For a $k$-option MCQ, we only make one change. We collect residual stream activations and train the knowledge and prediction probes as two single-layer **$k$-class** classifiers rather than binary classifiers. Specifically, given a hidden state  $h \in \mathbb{R}^{d \times 1}$,  $U_{\text{knowledge}}^{\top} h + b_{\text{knowledge}}$ and $U_{\text{prediction}}^{\top} h  + b_{\text{prediction}} \in \mathbb{R}^{k}$, where $U_{\text{knowledge}}, U_{\text{prediction}} \in \mathbb{R}^{d \times k}$ are weight matrices and $b_{\text{knowledge}}, b_{\text{prediction}} \in \mathbb{R}^{k \times 1}$ are biases, represent the logits for how likely each option is to be correct (knowledge) and how likely the model is to choose each option (prediction), respectively.
> These logits can be interpreted as coordinates in two $k$-dimensional subspaces of the residual stream: the knowledge subspace $\mathrm{span}(U_{\text{knowledge}})$ and the prediction subspace $\mathrm{span}(U_{\text{prediction}})$.
>
> **Intervention.** Following the same principle as scalar coordinate alignment in the binary case, the multiple-choice setting extends this procedure by aligning the $k$-dimensional logits (rather than one-dimensional logits in the binary setting) produced by the knowledge and prediction probes. This can be interpreted as aligning each hidden state's coordinates in the prediction subspace with those in the knowledge subspace. To calibrate hidden states jointly along all $k$ options, we solve the generalized version of the constrained optimization problem as in the binary case. This yields the generalized affine update:
> $$\tilde{h}' = \tilde{h} + \tilde{U}\_{\text{prediction}} (\tilde{U}\_{\text{prediction}}^{\top}\tilde{U}\_{\text{prediction}})^{-1} (\tilde{U}\_{\text{knowledge}}^{\top}\tilde{h} - \tilde{U}\_{\text{prediction}}^{\top}\tilde{h}),$$
> which reduces to the binary case when $k=1$.
> The update modifies the representation only within the prediction subspace, adjusting it to match the geometry of the representation in the knowledge subspace.
>
> **Experiments.** We extend our evaluation to standard MCQ benchmarks with more than two options. Specifically, we use the 4-choice formulation of BBH (BBH-4C), the original 3-option ARC-Challenge (ARC-Challenge-3C), and the original 4-option MMLU (MMLU-4C).
>
> As shown in Table1 below, under this multiple-choice setting with increased difficulty and more complex geometry, the Llama-2-7B base model shows a substantial gap between the prediction accuracy and the knowledge probe’s accuracy. However, KAPPA **consistently reduces the gap**, indicating that KAPPA generalizes well beyond binary-choice to multiple-choice settings. Notably, on Llama-2, KAPPA yields substantial gains, improving **BBH-4C** accuracy by nearly **+10%** and **MMLU-4C** accuracy by nearly **+5%**. For Qwen-2.5-7B, the improvements are smaller, which is expected because the original knowledge-prediction gap of the base model is smaller. Nevertheless, KAPPA effectively reduces the gap further. These results suggest that different models exhibit varying degrees of knowledge-prediction gap, while KAPPA consistently reduces it even in challenging multiple-choice scenarios.
>
> *Table1: Results of Multiple-Choice Intervention on BBH-4C, MMLU-4C, and ARC-3C using Llama-2 7B Chat and Qwen2.5 7B Instruct. Numbers represent test accuracy percentages. KAPPA variants correspond to interventions applied to 1, 3 layers, respectively.*
> | Method          | Llama-2 BBH | Llama-2 MMLU | Llama-2 ARC | Qwen2.5 BBH | Qwen2.5 MMLU | Qwen2.5 ARC |
> |-----------------|--------------|--------------|--------------|--------------|----------------|--------------|
> | Base            | 27.6         | 38.8         | 62.5         | 53.0         | 69.3           | 89.5         |
> | KAPPA (1-layer) | 34.1         | 43.9           | 64.4         | 56.1         | 69.9           | 89.7         |
> | KAPPA (3-layer) | 36.9         | 44.5         | 63.8         | 55.4         | 69.9           | 89.6         |
> | Knowledge Probe | 44.4         | 45.2         | 64.5         | 57.7         | 70.3           | 89.7         |

---

> ### Author Response · Authors · 2025-12-03
> **Official Response to Reviewer (Part 4 of 5)**
>
> >**Q1.** Could you explain more clearly what the "prediction basis" is? The description "encodes the probability of the model’s own choice" sounds like a tautology. More generally I find the anthropomorphic language used in the paper to describe the models makes understanding more difficult.
>
> **A1.** Thank you for pointing out that our explanation of the prediction basis was not clear. We have described **how to obtain the prediction basis** in our answer to *W1-A1* (also Section 3.1 of the uploaded revision). Below we **clarify why its weight vectors are called the prediction basis**, and we have incorporated this into the method section of the revision.
>
> Once trained, the weight vector ($u_{\text{prediction}}$) can be viewed as defining an axis in a subspace of the LLM’s activation space. For any hidden state (h), the scalar projection onto this axis ($u_{\text{prediction}}^\top h$) indicates the probability (its logit to be more precise) of the model's generated prediction (option A or B) indicated by the model's hidden state. In particular, a larger value of $u_{\text{prediction}}^\top h$ corresponds to the LLM's higher likelihood of predicting option “A”. This is why we call this axis the "prediction basis". We have added this explanation in the revised manuscript in introduction and method.
>
> Regarding your point on anthropomorphic language, the terms “knowledge basis”, “prediction basis”, and “knowledge-prediction subspace” are mathematically defined. While we have made our best effort to avoid anthropomorphic expressions, if the Area Chair identifies any that we missed, we will revise them accordingly.
>
> ---
>
> >**Q2.** In the intro line 47 you make the claim "We find that in the residual streams of certain layers, there exists a subspace spanned by two functionally distinct bases", can you explain how this proves there exists this subspace. And do you mean it exists in the LLM? The probes? or as a platonic form?
>
> **A2.** This question again is related to our answer in W1 and Section 3.1. Below, we clarify that **the subspace exists within the model’s residual stream hidden space**, and it can **always be defined as the span** of the knowledge and prediction directions.
>
> As we mentioned, in binary-choice settings, the knowledge probe ($u_{\text{knowledge}}$) encodes the probability of true answer indicated by the model's hidden state and the prediction probe ($u_{\text{prediction}}$) encodes the probability of model prediction (A or B) indicated by the model's hidden state with high accuracy. Since these probes are two directions within the model's hidden space, we can think of the 2D subspace formed by these two directions as axes (or bases) as a subspace that reliably captures the model's knowledge and prediction. In light of this, we answer your questions directly below.
>
> **"can you explain how this proves there exists this subspace?"**
>
> The subspace spanned by $u_{\text{knowledge}}$ and $u_{\text{prediction}}$ always exists. We define this subspace as the knowledge-prediction space because the two axes capture the model's knowledge and prediction.
>
> **"And do you mean it exists in the LLM? The probes? or as a platonic form?"**
>
> This subspace exists within the model's residual hidden space. It is not a platonic form; rather this subspace is precisely defined in mathematical terms. The model's residual hidden states constitute a d-dimensional space, and the referred subspace is a subspace of this d-dimensional space spanned by two axes $u_{\text{knowledge}}$ and $u_{\text{prediction}}$ obtained through probing.
>
> ---
>
> >**Q3.** Can this be interpreted as a very low rank LoRA?
>
> **A3.** We thank the reviewer for this insightful observation. As noted in our response to Q1, our method belongs to the inference-time intervention paradigm rather than fine-tuning and modifying model parameters. Nevertheless, we agree that the formulation of KAPPA can indeed be written as a rank-1 linear mapping, which is **mathematically similar** to the structure of a LoRA update.
>
> Below, we clarify **(1) how our method can be rewritten** in a form that is mathematically similar to a rank-1 affine transform, and **(2) why KAPPA nevertheless fundamentally differs from LoRA** in its operation.
>
> **(1) Rewriting our method as a Rank-1 Affine Transformation**
>
> Starting from Eq.~2 in the paper, we can rewrite the transformation as $h' = h + BAh + C$, where
> $B = \frac{u_{\text{pred}}}{u_{\text{pred}}^\top u_{\text{pred}}} \in \mathbb{R}^{d \times 1}$,
> $A = (u_{\text{know}} - u_{\text{pred}})^\top \in \mathbb{R}^{1 \times d}$,
> $C = u_{\text{pred}} \cdot \frac{b_{\text{know}} - b_{\text{pred}}} {u_{\text{pred}}^\top u_{\text{pred}}} \in \mathbb{R}^{d \times 1}$.

---

> ### Author Response · Authors · 2025-12-03
> **Official Response to Reviewer (Part 5 of 5)**
>
> **Q3-A3(continued)**
>
> Here, the matrix (BA) is rank-1, and the bias term (C) is typically very small in practice. Thus, the resulting affine transformation resembles a rank-1 LoRA update in form. Although we did not originally interpret our method through this lens, we appreciate this perspective and have incorporated it into the revision.
>
> **(2) Differences between KAPPA and LoRA**
>
> Despite KAPPA's similar form to LoRA, we further clarify below how KAPPA differs from LoRA in its operation.
> - **Input/output:** LoRA adapters receive the input to each linear layer and modify the layer’s output. In contrast, KAPPA takes the residual-stream activation as input and applies a transformation directly to the hidden state at inference time.
> - **Mechanism:** LoRA adds a low-rank update to the existing weight matrix, altering model parameters through training. KAPPA leaves all weights untouched and applies an inference-time correction to the residual-stream activations.
> - **Parameter scale and role:** The LoRA baseline attaches adapters to every linear weight across all layers to compress new task-specific information. KAPPA intervenes on only a single layer, thus using much fewer parameters ($32 \times 16 \times 4$ times fewer for Llama-2; $28 \times 16 \times 4$ times fewer for Qwen) These parameters serve solely to ensure that the model’s existing knowledge is faithfully expressed in its predictions, not to encode new knowledge.

---

### Author Response · Authors · 2025-12-03
**General Response to All Reviewers (Part 2 of 2)**

In response to the reviewers’ **concerns**, we provided **clarifications** and strengthened our presentation as summarized below:

- Reviewer ePcc asked how our approach differs from fine-tuning. We clarified that, unlike fine-tuning—which updates model parameters to inject new knowledge—our method freezes all weights and applies an inference-time intervention to better express knowledge already encoded in the model. Furthermore, our main goal is to provide a geometric lens to better understand the knowledge-prediction gap and reduce the gap, rather than to improve the model’s task accuracy.
- Reviewers ePcc and 2gxg questioned the practical strength of our approach. We explained that our method allows the model to make predictions that are faithful to its internal knowledge, even without fine-tuning the model. This often improves task accuracy as well, as evidenced across our experiments. The simple geometric correction reliably achieves this goal from multiple-choice questions to free-form questions. In addition, we highlight that our main contribution is an interpretable geometric framework for understanding and manipulating the knowledge–prediction relationship.
- We have substantially improved clarity and presentation throughout the paper. Specifically, we:
  - (i) added a clearer explanation of our methodology (ePcc)
  - (ii) included missing references (jtgY)
  - (iii) clarified dataset and model details (HJVi)
  - (iv) corrected formatting issues regarding image size and captions (HJVi)

---

### Author Response · Authors · 2025-12-03
**General Response to All Reviewers (Part 1 of 2)**

We sincerely appreciate all reviewers for their thoughtful feedback and constructive suggestions on our manuscript. Below, we summarize the **key contributions** of our work using **quotes** from the reviewers.


- Intuitive and promising geometric view of the knowledge-prediction gap in LLMs
  - “The paper provides **a clear intuitive framework** for understanding the knowledge-prediction gap through a two-dimensional subspace representation” (jtgY)
  - “**A promising insight into the geometry of the model's "knowledge"** in its internal representations.” (HJVi)
  - “I think the steering intervention as a correction in a learned plane is **neat and should be explored more.**” (ePcc)

- Strong motivation and mathematically sound methods
  - “The **motivation** behind the method in general, as well as many of its specific design choices, **is clear.**” (2gxg)
  - “An i**nteresting intervention** method with **well-developed mathematical background.**” (HJVi)
  - “The analysis and methods look **sound** ...” (ePcc)

- Extensive experiments and performance improvements
  - “The authors conduct **extensive experiments** …”, “The method **improves LLMs performance** across several datasets and extends to free-form generation.” (jtgY)
  - “Results show that the proposed method **noticeably improves the model's performance.**” (HJVi)

Reviewers suggested **two additional experiments,** and we conducted all of them thoroughly and incorporated into our revision as follows:

#### **1. Generalization to Multiple-Choice Questions (ePcc, jtgY, HJVi, 2gxg):**
All reviewers consistently raised the question of whether our approach generalizes to multiple-choice settings with more than two choices. To address this, we **extended our initially binary-choice framework to general multiple-choice cases** (detailed below) and conducted **new experiments on three standard MCQ benchmarks** (BBH, MMLU, ARC-Challenge with **3**- and **4**-choice questions) across two model families (Llama-2 and Qwen-2.5 7B).

The generalized method yields **consistent improvements** across all datasets and models, including a notable **+10% accuracy** on BBH (4-choice) and **+5% accuracy** on MMLU (4-choice) for Llama-2-7B. These results show that our method effectively identifies the knowledge-prediction gap even in challenging multiple-choice settings and effectively reduces the gap.

**Details about the extension:** The extension from binary-choice to any k-choice questions is very straightforward. The only change we make is to train the knowledge and prediction probes as two single-layer **$k$-class classifiers** (instead of binary classifiers). These two probes can then be used to compute the k-dimensional logits that estimate (i) the likelihood that each option is the correct answer (knowledge) and (ii) the likelihood that the LLM will select each option (prediction), respectively. Our method then aligns the two k-dimensional logits (instead of a single logit for the binary-choice setting). This corresponds to aligning each hidden state's coordinates in the k-dimensional prediction subspace with those in the knowledge subspace, spanned respectively by the two probe weight matrices. This extension is a **straightforward generalization** of the original update: it follows directly from the same closed-form optimization used in the binary case and reduces exactly to the original formulation when $k=1$.

#### **2. Additional Experiments on More Models (jtgY, HJVi):**
Reviewers encouraged evaluating a wider range of LLMs beyond the originally reported Llama-2 and Qwen2.5 7B models. In response, we expanded our evaluation to **Qwen3-32B, Llama-3.1-8B,** and **Qwen3-4B** on the BBH-binary dataset. Across all three models, we observe **a clear knowledge-prediction gap that persists even in larger and more recent architectures**, and our method **consistently improves accuracy** by **+9.1%p**, **+13.3%p**, and **+13.7%p**, respectively. These results highlight that our method reliably closes the pervasive gap across diverse model sizes and families.

---

### Meta-Review · Area_Chair_b4nL · 2025-12-31

**Summary:**

The paper investigates the knowledge-prediction gap, a phenomenon where LLMs possess the internal knowledge necessary to answer a question correctly (extractable via probing) but fail to select the correct answer in a multiple-choice format. The authors propose KAPPA, an inference-time intervention that identifies a 2D subspace in the residual stream and aligns the prediction basis with the knowledge basis to make model outputs more faithful to their latent knowledge.

The main concerns focus on:
- The original submission focused almost exclusively on binary-choice questions. and did not provide deeper analysis for multiple-choice questions, which are more common in the real world situation
- Concerns regarding whether the method works on larger, newer models (>7B parameters)
- Questions regarding how this differs from LoRA and missing references to prior work on internal knowledge representation

**Reviewer Concerns:**

Addressed concerns:
- The authors added evaluations on larger models (32B) and more recent architectures (Llama-3.1 and Qwen 3). The gap was shown to persist across scales, and KAPPA consistently reduced it
- Authors added new discussion compared to the missing related work

Outstanding concerns:
- Although authors provided new experiments about multiple-choice questions setting, it lacks deeper analysis and explanations from the geometric space from the model internally, and how different those new setting compared to the binary-choice. In addition, the paper title and abstract are misleading due to this missing deeper analysis about multiple-choice questions settings.
- While authors argued that KAPPA is an inference-time intervention which does not need parameter updates, it is partially unfair to compare with LoRA, as LoRA is used to fine tune the whole adapter, but KAPPA just modify few parameters. Instead, this work should compare other inference-time intervention methods.

**Reviewer Scores:**

Due to multiple reviewers' main concerns about the binary-choice setting without deeper analysis on the multiple-choice questions, this outstanding concerns lead to different reviewers to probably maintain their scores. In addition, the comparison with LoRA rather than other inference-time intervention method lead to the score unchanged.

---

### Decision · Program_Chairs · 2026-01-26

Reject